# Nanos promotes epigenetic reprograming of the germline by down-regulation of the THAP transcription factor LIN-15B

**Chih-Yung Sean Lee, Tu Lu, Geraldine Seydoux***

Department of Molecular Biology and Genetics, Howard Hughes Medical Institute, Johns Hopkins University School of Medicine, Baltimore, United States

**Abstract** Nanos RNA-binding proteins are required for germline development in metazoans, but the underlying mechanisms remain poorly understood. We have profiled the transcriptome of primordial germ cells (PGCs) lacking the *nanos* homologs *nos-1* and *nos-2* in *C. elegans*. *nos-1nos-2* PGCs fail to silence hundreds of transcripts normally expressed in oocytes. We find that this misregulation is due to both delayed turnover of maternal transcripts and inappropriate transcriptional activation. The latter appears to be an indirect consequence of delayed turnover of the maternally-inherited transcription factor LIN-15B, a synMuvB class transcription factor known to antagonize PRC2 activity. PRC2 is required for chromatin reprogramming in the germline, and the transcriptome of PGCs lacking PRC2 resembles that of *nos-1nos-2* PGCs. Loss of maternal LIN-15B restores fertility to *nos-1nos-2* mutants. These findings suggest that Nanos promotes germ cell fate by downregulating maternal RNAs and proteins that would otherwise interfere with PRC2-dependent reprogramming of PGC chromatin.

DOI: https://doi.org/10.7554/eLife.30201.001

## Introduction

In animals, formation of the germline begins during embryogenesis when a few cells (~30 in mice, two in *C. elegans*) become fated as primordial germ cells (PGCs) – the founder cells of the germline. PGC specification requires the activity of chromatin regulators that induce genome-wide changes in gene expression. For example, in mice, the transcriptional repressor BLIMP1 initiates PGC specification by blocking the expression of a mesodermal program active in neighboring somatic cells (*Ohinata et al., 2005*; *Saitou et al., 2005*). In *C. elegans*, the NSD methyltransferase MES-4 and the Polycomb Repressive Complex (PRC2, including MES-2, 3 and 6) cooperate to place active and repressive histone marks on germline and somatic genes, respectively (*Gaydos et al., 2012*). Despite their critical roles during germ cell development, the BLIMP1 and MES/PRC2 chromatin regulators are not germline-specific factors and also function during the differentiation of somatic lineages (*Cui et al., 2006*; *Gaydos et al., 2012*; *Seydoux and Braun, 2006*). How the activities of these global regulators are modulated in germ cells to promote a germline-specific program is not well understood.

In *C. elegans*, genetic analyses have shown that MES-dependent activation of germline genes is antagonized in somatic lineages by a group of transcriptional regulators first identified by their effects on vulval development (*Curran et al., 2009*; *Petrella et al., 2011*; *Unhavaithaya et al., 2002*). Among these, components of the DRM (named for its Dp/E2F, pRB, and MuvB subunits) class of transcriptional regulators and LIN-15B, a THAP domain DNA binding protein, have been implicated in the silencing of germline genes in somatic cells (*Petrella et al., 2011*; *Wu et al., 2012*). Loss of DRM factors or LIN-15B causes ectopic activation of germline genes in somatic cells leading to growth arrest at elevated temperatures (26°C). Inactivation of *mes-2*, *mes-3*, *mes-4* and *mes-6*

**\*For correspondence:** gseydoux@jhmi.edu

**Competing interests:** The authors declare that no competing interests exist.

**eLife digest** Every new embryo inherits a set of starting instructions from its mother. These instructions are called a 'maternal dowry' and help a fertilized egg through the first few stages of development. Later, the maternal dowry is removed so that the embryo's genetic instructions can take over.

In animals, some of the cells in this early embryo become specialized to produce eggs (technically called oocytes) or sperm. These cells are called germ cells, and they are needed for reproduction. A protein called Nanos helps germ cells become different to other cells, but it is not clear how Nanos has this effect.

Lee et al. studied Nanos in the embryos of the worm *Caenorhabditis elegans*. In many ways, early development is the same in the worm as in many other animals. By examining worms that did not have Nanos, Lee et al. showed that germ cells without Nanos do not lose their maternal dowry. As a result, the cells still contain a molecule called LIN-15B, which makes other types of cells in the worm. Ultimately, without Nanos, the germ cells do not develop and die leaving the worm sterile.

Germ cells are essential for living things to reproduce and have children. Understanding how they are created can teach scientists a lot about how embryos develop before birth. This could eventually help to boost fertility in endangered species or to treat human sterility.

DOI: https://doi.org/10.7554/eLife.30201.002

suppresses the ectopic germline gene expression and restores viability to *lin-15B* mutants at 26°C (*Petrella et al., 2011*). These observations have suggested that DRM factors and LIN-15B antagonizes MES activity in somatic lineages to keep germline genes off (*Petrella et al., 2011*). A similar antagonism, but in reverse, has been uncovered in the adult germline between the NSD methyltransferase MES-4 and the DRM transcription factor LIN-54 (*Tabuchi et al., 2014*). The X chromosome is a major focus of MES repression in *C. elegans* germline. The X chromosome is silenced throughout germ cell development except in oocytes, which activate the transcription of many X-linked genes (*Kelly et al., 2002*). *mes* mutants prematurely activate the transcription of X-linked genes in pregametic germ cells leading to germ cell death (*Bender et al., 2006*; *Gaydos et al., 2012*; *Seelk et al., 2016*). Reducing the function of the synMuvB transcription factor *lin-54* in *mes-4* mutants restores the expression of X-linked genes closer to wild-type levels (*Tabuchi et al., 2014*). Therefore in the germline, MES activity antagonizes DRM activity to keep the X chromosome silent. Together, these genetic studies suggest that competition between the MES chromatin modifiers and the DRM/LIN-15B transcription factors balance the transcription of somatic and germline genes in somatic and germline tissues. How this competition is biased during development to ensure appropriate gene expression in each tissue is not known. In this study, we have discovered a link between Nanos and LIN-15B that provides an explanation for how MES activity becomes dominant in the nascent germline (*Figure 1A*).

The *C. elegans* PGCs arise early in embryogenesis from pluripotent progenitors (P blastomeres) that also generate somatic lineages (*Figure 1B*). RNA polymerase II activity is repressed in the P lineage until the 100 cell stage when the last P blastomere $P_4$ divides to generate Z2 and Z3, the two PGCs (*Seydoux et al., 1996*). RNA polymerase II becomes active in PGCs, but these cells remain relatively transcriptionally quiescent, and exhibit reduced levels of active chromatin marks compared to somatic cells throughout the remainder of embryogenesis (*Kelly, 2014*). Active marks and robust transcription return after hatching when the L1 larva begins to feed and the PGCs resume proliferation in the somatic gonad (*Fukuyama et al., 2006*; *Kelly, 2014*). The mechanisms that maintain PGC chromatin in a silenced state during embryogenesis are not known, but embryos lacking the *nanos* homologs *nos-1* and *nos-2* have been reported to display abnormally high levels of the active mark H3meK4 mark in PGCs (*Schaner et al., 2003*). *nos-1nos-2* PGCs initiate proliferation prematurely during embryogenesis and die during the second larval stage (*Subramaniam and Seydoux, 1999*). Nanos proteins are broadly conserved across metazoans and have been shown to be required for PGC survival in several phyla, from insects to mammals (*Asaoka-Taguchi et al., 1999*; *Beer and Draper, 2013*; *Deshpande et al., 1999a*; *Lai et al., 2012a*; *Tsuda et al., 2003*). Nanos proteins are cytoplasmic RNA-binding proteins that regulate gene expression post-transcriptionally by recruiting

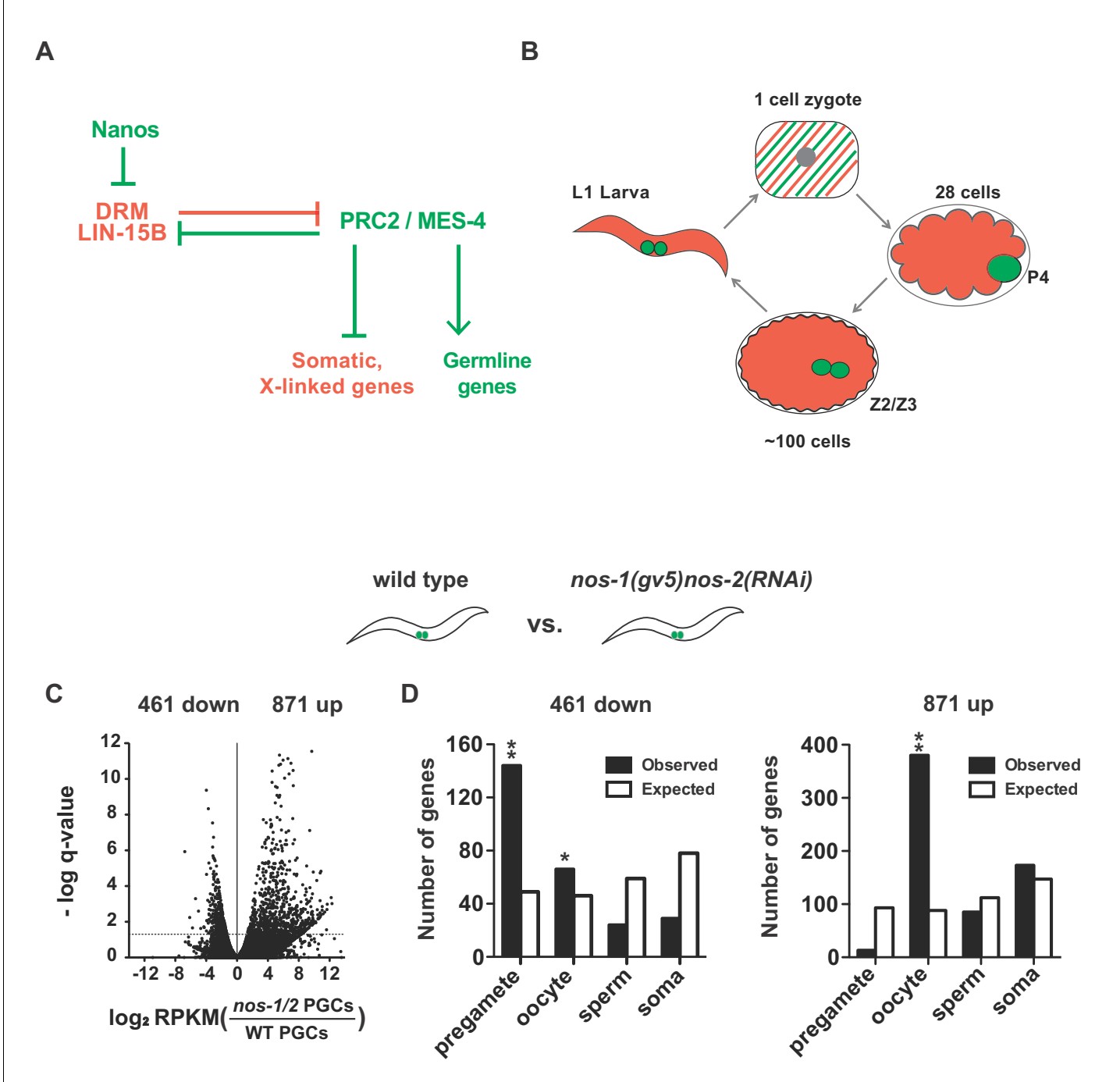

Figure 1. *nos-1nos-2* PGCs upregulate transcripts expressed in oocytes. (A) Mutual antagonism between DRM/LIN-15B transcription factor and PRC2/ MES-4 chromatin modifiers balances activities that promote somatic (red) and germline (green) gene expression. In somatic lineages, LIN-15B and DRM transcription factors opposes PRC2 to silence germline genes (*Petrella et al., 2011*). In pregametic germ cells, PRC2 activates germline genes (with the help of MES-4) and silences somatic genes and X-linked genes, including genes expressed in oocytes (e.g. *lin-15B*) (*Gaydos et al., 2012*). In this study, we show that Nanos is required to remove maternally-inherited LIN-15B from PGCs to allow proper PRC2/MES-4 function. (B) Germline cycle in *C. elegans*. The zygote (stripped red and green) inherits from the oocyte maternal mRNAs that promote the development of somatic (red) and germline (green) cell fates during embryogenesis. $P_4$ is the germline founder cell that gives rise to Z2 and Z3, the two primordial germ cells. Z2 and Z3 do not divide and remain mostly transcriptionally quiescent during embryogenesis. They resume division and transcription in the first larval stage (L1). (C–D). Transcriptome comparison between PGCs isolated from wild-type and *nos-1(gv5)nos-2(RNAi)* L1 larvae using SMART-seq libraries (Materials and methods, See *Figure 1—figure supplement 1B–C* for results with Truseq libraries). (C) Volcano plot showing log2 fold-change in transcript abundance for each gene. The number of genes that were significantly up or downregulated in *nos-1(gv5)nos-2(RNAi)* (designated as *nos-1/2*) PGCs are indicated. *Figure 1 continued on next page*

*Figure 1 continued*

Dashed line marks the significance cutoff of q = 0.05 (Y axis) above which genes were counted as misexpressed. (**D**) Bar graphs showing expected and observed number of genes (Y axis) in different expression categories (X axis). Genes were assigned to a particular expression category based on their preferential expression patterns as determined in (*Gaydos et al., 2012*; *Ortiz et al., 2014*); Materials and methods and *Supplementary file 1*). The lists are non-overlapping and include 1694 pregamete genes, 1594 oocyte genes, 2042 sperm genes, and 2684 somatic genes. Oocyte genes include genes required for oogenesis and maternal genes required for embryogenesis. Because genes were categorized based on their preferential gene expression pattern, genes on one list may also be expressed in other tissues. See *Supplementary file 1* for complete gene lists. Pregamete and oocyte genes are overrepresented among downregulated genes and oocyte genes are overrepresented among upregulated genes. Asterisks indicate significantly more genes than expected (hypergeometric test, p-value<0.01 [*] or <0.001 [**]).

DOI: https://doi.org/10.7554/eLife.30201.003

The following figure supplement is available for figure 1:

**Figure supplement 1.** *nos-1nos-2* PGCs upregulate transcripts expressed in oocytes.

DOI: https://doi.org/10.7554/eLife.30201.004

effector complexes that silence and degrade mRNAs in the cytoplasm. Six direct Nanos mRNA targets have been identified to date [Drosophila *hunchback, cyclin B and hid* (*Asaoka-Taguchi et al., 1999*; *Dalby and Glover, 1993*; *Kadyrova et al., 2007*; *Murata and Wharton, 1995*; *Sato et al., 2007*; *Wreden et al., 1997*), Xenopus *VegT* (*Lai et al., 2012a*), and sea urchin *CNOT6* and *eEF1A* (*Oulhen et al., 2017*; *Swartz et al., 2014*)]], but none of these targets are sufficient to explain how Nanos activity might affect PGC chromatin. In this study, we characterize the gene expression defects of PGCs lacking *nanos* activity in *C. elegans*. Our findings indicate that *nanos* activity is required in PGCs to silence maternal transcripts inherited from the oocyte. We identify maternal *lin-15B* as a critical target of Nanos regulation that must be turned-over to establish MES dominance in PGCs.

## Results

### PGCs lacking *nos-1* and *nos-2* upregulate oocyte transcripts

*nos-2* is provided maternally and functions redundantly with zygotically-expressed *nos-1* (*Subramaniam and Seydoux, 1999*). To generate large numbers of larvae lacking both *nos-1* and *nos-2* activities, we fed hermaphrodites homozygous for a deletion in *nos-1* [*nos-1(gv5)*] bacteria expressing *nos-2* dsRNA and collected their progenies at the L1 stage [hereafter designated *nos-1(gv5)nos-2(RNAi)* L1 larvae]. We used fluorescence-activated cell sorting (FACS) to isolate PGCs based on expression of the germ cell marker PGL-1::GFP and processed the sorted cells for RNA-seq (L1 PGCs). Two independent RNA-seq libraries (biological replicates) were analyzed for each genotype (wild-type and *nos-1(gv5)nos-2(RNAi)*) using Tophat 2.0.8 and Cufflinks 2.0.2 software (*Trapnell et al., 2012*). These analyses identified 461 underexpressed transcripts and 871 overexpressed transcripts in *nos-1(gv5)nos-2(RNAi)* L1 PGCs compared to wild-type (q < 0.05, *Figure 1C* and *Supplementary file 5* for list of misregulated genes). qRT-PCR of 11 genes confirmed the result of the RNA-seq analysis (*Figure 1—figure supplement 1A*).

To determine the types of genes affected, we used published gene expression data (*Gaydos et al., 2012*; *Meissner et al., 2009*; *Ortiz et al., 2014*; *Reinke et al., 2004*; *Wang et al., 2009*) to generate non-overlapping gene lists with preferential expression in pregametic germ cells (germline stem cells and early meiotic cells), oocytes, sperms, or somatic cells (Materials and methods and *Supplementary file 1*). The oocyte list includes both genes required for oogenesis and maternal genes required for embryonic development. We found that 31% (144/461) of underexpressed transcripts in *nos-1(gv5)nos-2(RNAi)* L1 PGCs correspond to genes expressed preferentially in pregametic germ cells (*Figure 1D*). These include *sygl-1*, a gene transcribed in germline stem cells in response to Notch signaling from the somatic gonad (*Kershner et al., 2014*). The *sygl-1* transcript was decreased by 4.7-fold in *nos-1(gv5)nos-2(RNAi)* PGCs (*Supplementary file 4*). In contrast, overexpressed transcripts in *nos-1(gv5)nos-2(RNAi)* L1 PGCs correspond primarily to genes expressed in oocytes (380/871) (*Figure 1D*). These include *lin-41*, a master regulator of oocyte fate (*Spike et al., 2014a*; *2014b*) and *tbx-2* and *hnd-1*, transcription factors that promote muscle development during embryogenesis(*Fukushige et al., 2006*; *Smith and Mango, 2007*). The *lin-41*,

*tbx-2,* and *hnd-1* transcripts were upregulated 5.1-fold, 11-fold and 91-fold, respectively, in *nos-1 (gv5)nos-2(RNAi)* PGCs (*Supplementary file 4*). We conclude that *nos-1(gv5)nos-2(RNAi)* PGCs overexpress oogenic and maternal genes normally expressed in oocytes, and fail to activate pregametic genes normally expressed in PGCs.

## Turnover of maternal transcripts is delayed in PGCs lacking *nos-1* and *nos-2*

The overexpressed oocyte-class transcripts in *nos-1(gv5)nos-2(RNAi)* L1 PGCs could correspond to maternal transcripts that failed to turnover during embryogenesis or to zygotic transcripts synthesized de novo in *nos-1(gv5)nos-2(RNAi)* PGCs. To distinguish between these possibilities, we first examined the fate of maternal RNAs in PGCs. We isolated PGCs from embryos with fewer than 200 cells, at a time when PGCs are still mostly transcriptionally silent (EMB PGCs) (*Schaner et al., 2003*; *Seydoux and Dunn, 1997*). By comparing the EMB PGC transcriptome to a published oocyte transcriptome (*Stoeckius et al., 2014*), we observed an excellent correlation in relative transcript abundance between oocytes and EMB PGCs (*Figure 2—figure supplement 1A*), suggesting that the transcriptome of EMB PGCs consists primarily of maternal mRNAs inherited from the oocyte. This finding is consistent with in situ hybridization experiments that showed that many maternal RNAs persist in the embryonic germ lineage at least to the $P_4$ germline founder cell (*Seydoux and Fire, 1994*). Next, we compared the transcriptome of EMB PGCs to that of L1 PGCs to identify PGC transcripts whose abundance decline during embryogenesis. We identified 411 down-regulated transcripts, including 197 oocyte transcripts (*Figure 2A and B* and *Supplementary file 5*), consistent with turnover of many maternal mRNAs in PGCs after the 200 cell stage. Strikingly, the amplitude of this turnover was diminished in *nos-1(gv5)nos-2(RNAi)* mutants: the abundance of the 411 transcripts remained high overall during the transition from EMB PGCs to L1 PGCs in *nos-2(RNAi)nos-1(gv5)* embryos (*Figure 2C*). Furthermore, when comparing wild type and *nos-1(gv5)nos-2(RNAi)* EMB PGCs, we identified 182 differentially expressed transcripts (11 down- and 171 upregulated), including 71 of oocyte transcripts that were more abundant in *nos-1(gv5)nos-2(RNAi)* EMB PGCs (*Figure 2—figure supplement 1B*). Together these findings suggest a defect in maternal mRNA turnover in *nos-1(gv5)nos-2(RNAi)* PGCs that is already detectable at the 200 cell stage and persists through embryogenesis.

To test this hypothesis directly, we performed in situ hybridization experiments against three maternal mRNAs. In wild-type embryos, *mex-5*, C01G8.1 and Y51F10.2 are turned over rapidly in somatic lineages (before the 28 cell stage) and more slowly in the germ lineage (200–300 cell stage for *mex-5* and C01G8.1; bean-stage for Y51F10.2). We found that, in *nos-1(gv5)nos-2(ax3103)* embryos, turnover was not affected in somatic lineages, but was delayed in PGCs, with C01G8.1 and *mex-5* persisting to the ~500 cell stage and Y51F10.2 persisting to 1.5-fold stage (*Figure 2D*). We conclude that turnover of maternal mRNAs is compromised in *nos-1(gv5)nos-2(RNAi)* PGCs.

## PGCs lacking *nos-1* and *nos-2* activate the transcription of many genes normally silent in PGCs

By the L1 stage, PGCs are transcriptionally active. To explore whether inappropriate transcription also occurs in *nos-1nos-2* PGCs by the L1 stage, we examined transcripts that increase in abundance during the transition from EMB and L1 PGCs. We identified 130 such transcripts in wild-type PGCs, including 30% in the pregamete category (39/130, *Figure 2B*), consistent with PGCs transitioning to a pregamete fate by the L1 stage. In contrast, in *nos-1(gv5)nos-2(RNAi)* PGCs, many more (510) transcripts increased in abundance during embryogenesis, and these were distributed among all genes categories, including oocyte genes (16%, 84/510 *Figure 2E and F* and *Supplementary file 5*). This finding suggests that, unlike wild-type PGCs, *nos-1(gv5)nos-2(RNAi)* PGCs fail to transition to a pregamete program and instead adopt a hybrid transcriptional profile that includes activation of oocyte genes.

To explore this possibility further, we used ATAC-seq to identify regions of 'open' chromatin that differ between wild-type and *nos-1(gv5)nos-2(RNAi)* L1 PGCs (*Buenrostro et al., 2015*). We identified 221 genes that showed increased chromatin accessibility at their promoter region in *nos-1(gv5) nos-2(RNAi)* L1 PGCs compared to wild-type ('ATAC-seq+' genes; *Figure 3*, *Figure 3—figure supplement 1*, and *Supplementary file 2*). Consistent with transcriptional activation, most of the ATAC-

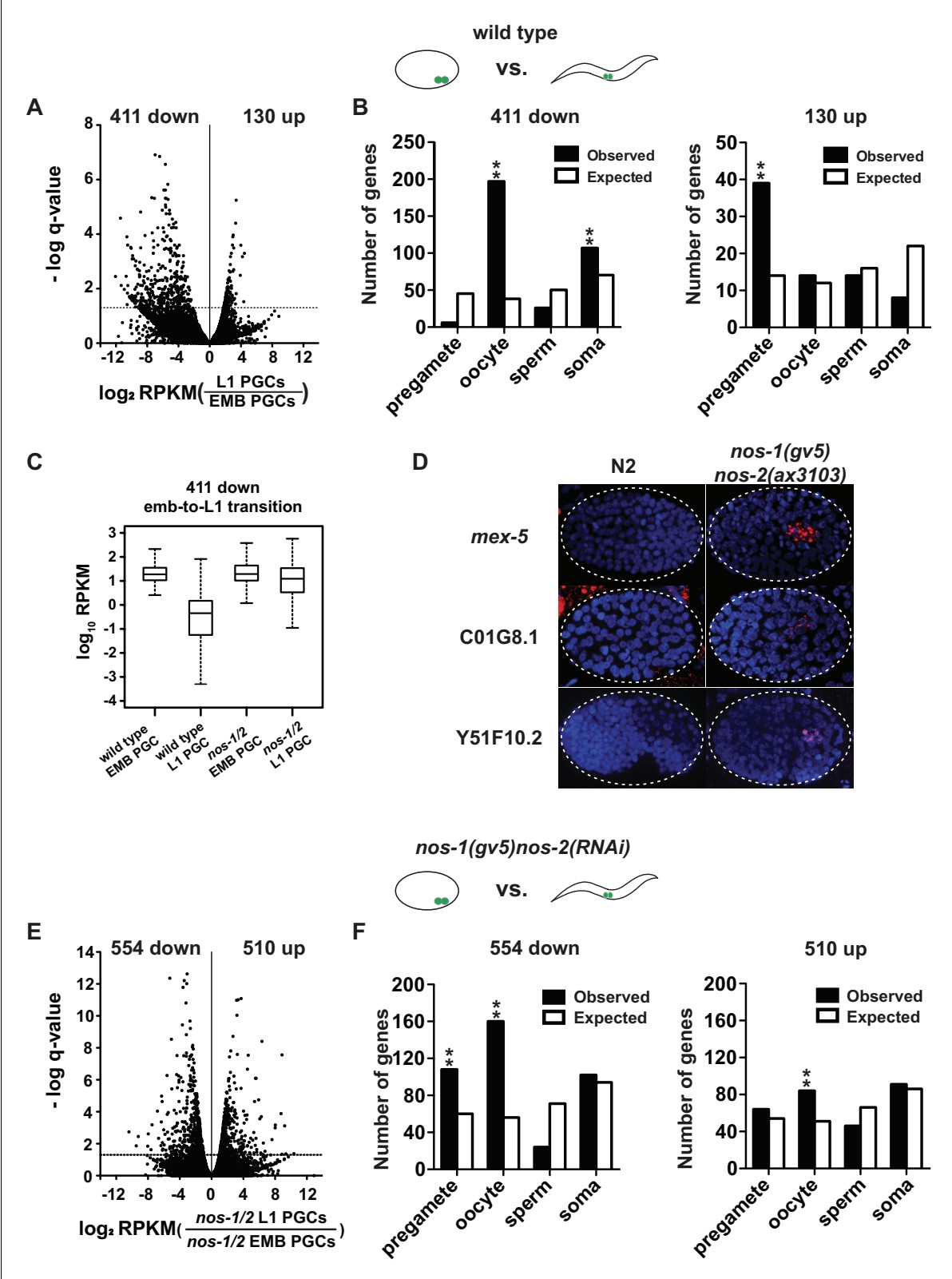

**Figure 2.** *nos-1nos-2* PGCs are defective in maternal mRNA turnover during embryogenesis. (A–B) Transcriptome comparison between PGCs isolated from wild-type embryos and wild-type L1 larvae. (A) Volcano plot showing log2 fold-change in transcript abundance for each gene. The numbers of genes whose expression were up or downregulated in L1 PGCs compared to embryonic PGCs are indicated. Dashed line marks the significance cutoff of q = 0.05 above which genes were counted as misexpressed. (B) Bar graphs showing expected and observed number of genes (Y axis) in the different

*Figure 2 continued on next page*

Figure 2 continued

expression categories (X axis). The lists of expression categories used here are the same as those in *Figure 1B* (*Supplementary file 1*). Oocyte and soma genes are overrepresented among downregulated genes and pregamete genes are overrepresented among upregulated genes. Asterisks indicate significantly more genes than expected (hypergeometric test, p-value<0.001 [**]). (C) Box and whisker plot showing the expression levels (log10) of 411 genes that are downregulated during embryogenesis in wild-type PGCs. Expression of these genes remains high on average in *nos-1 (gv5)nos-2(RNAi)* PGCs (designated as *nos-1/2*). Each box extends from the 25th to the 75th percentile, with the median indicated by the horizontal line; whiskers extend from the 2.5th to the 97.5th percentiles. (D) Photomicrograph of embryos hybridized with single molecule fluorescence probes (red) against *mex-5*, C01G8.1 and Y51F10.2. Wild-type and *nos-1(gv5)nos-2(ax3103)* embryos were raised at 25°C and are compared here at the same stage (as determined by the number of DAPI-stained nuclei shown in blue). By the stages shown, all three transcripts have turned over in wild-type (N2), but are still present (red signal) in PGCs in *nos-1(gv5)nos-2(ax3103)* embryos. At least 10 embryos were examined per probe set in different genotypes shown. (E–F) Transcriptome comparison between PGCs isolated from *nos-1(gv5)nos-2(RNAi)* embryos and *nos-1(gv5)nos-2(RNAi)* (designated as *nos-1/2*) L1 larvae. (E) Volcano plot showing log2 fold-change in transcript abundance for each gene. The numbers of genes whose expression were up or downregulated in L1 PGCs compared to embryonic PGCs are indicated. Dashed lines mark the significance cutoff of q = 0.05 above which genes were counted as misexpressed. (F) Bar graphs showing expected and observed number of genes (Y axis) in the different expression categories (X axis). Asterisks indicate significantly more genes than expected (hypergeometric test, p-value<0.001 [**]).

DOI: https://doi.org/10.7554/eLife.30201.005

The following source data and figure supplements are available for figure 2:

**Figure supplement 1.** Maternal RNAs are maintained in early embryonic PGCs.
DOI: https://doi.org/10.7554/eLife.30201.006

**Figure supplement 2.** Perdurance of maternal RNAs is not suppressed by loss of *lin-15B* in *nos-1(gv5)nos-2(ax3103)* PGCs.
DOI: https://doi.org/10.7554/eLife.30201.007

**Figure supplement 2—source data 1.** R code for comparing transcriptome between oocyte and embryonic blastomeres.
DOI: https://doi.org/10.7554/eLife.30201.008

**Figure supplement 3.** Principal component analysis (PCA) of transcriptomes from isolated PGCs.
DOI: https://doi.org/10.7554/eLife.30201.009

---

seq+ genes were overexpressed in *nos-1(gv5)nos-2(RNAi)* L1 PGCs compared to wild-type (*Figure 3A*). Furthermore, 108/221 of the ATAC-seq +genes were oocyte genes (*Figure 3B*). In contrast, we identified 29 genes with decreased chromatin accessibility in *nos-1(gv5)nos-2(RNAi)* compared to wild-type (*Figure 3—figure supplement 1* and *Supplementary file 2*), most of which (13/29) were in the pregametic category (*Figure 3—figure supplement 1*). These observations confirm that *nos-1nos-2* PGCs fail to fully activate the transcription of pregamete genes and instead activate many oocyte genes.

Transcription of the X chromosome is silenced in all germ cells except in oocytes (*Kelly et al., 2002*). If *nos-1nos-2* PGCs are adopting an oocyte-like transcriptional program, we would expect X-linked genes to be active. Strikingly, 63% (139/221) of the ATAC-seq+ genes were X-linked (*Figure 3C*). Furthermore, we found that, while transcripts from X-linked genes are rare in wild-type L1 PGCs (average 4.7 FPKM per X-linked genes compared to 50.9 for autosomal genes), they are more abundant in *nos-1(gv5)nos-2(RNAi)* L1 PGCs (9.6 FPKM for X-linked genes compared to 43.8 for autosomal genes) (*Supplementary file 3*). We conclude that silencing of the X chromosome is defective in *nos-1(gv5)nos-2(RNAi)* PGCs, consistent with these cells adopting an oocyte-like transcriptional profile.

## PGCs lacking *mes-2* or *mes-4* upregulate X-linked genes

Failure to silence X-linked genes has been reported for germ cells lacking the chromatin regulators *mes-2* and *mes-4* (*Bender et al., 2006*; *Gaydos et al., 2012*). To directly compare the effect of loss of *nos* versus *mes* function in PGCs, we purified PGCs from L1 larvae derived from hermaphrodites where *mes-2* or *mes-4* was inactivated by RNAi (Materials and methods). As expected, loss of *mes-2* and *mes-4* led to a significant upregulation of X-linked genes in L1 PGCs (*Figure 4A–B*, *Figure 4—figure supplement 2*, and *Supplementary file 5* for lists of misregulated genes). To directly compare these changes to those observed in *nos-1(gv5)nos-2(RNAi)* PGCs, we compared, for each genotype, the log2 fold change over wild-type for X-linked genes and for autosomal oocyte genes. As expected, we observed a strong positive correlation between *mes-2* and *mes-4* in both gene categories (R = 0.91 and R = 0.76, X-linked and autosomal oocyte genes, respectively) (*Figure 4C and D*). We also observed a strong correlation between *mes-4(RNAi)* and *nos-1(gv5)nos-2(RNAi)* (R = 0.75, *Figure 4E*, *Figure 4—figure supplement 1*) and *mes-2(RNAi)* and *nos-1(gv5)nos-2(RNAi)* (R = 0.73,

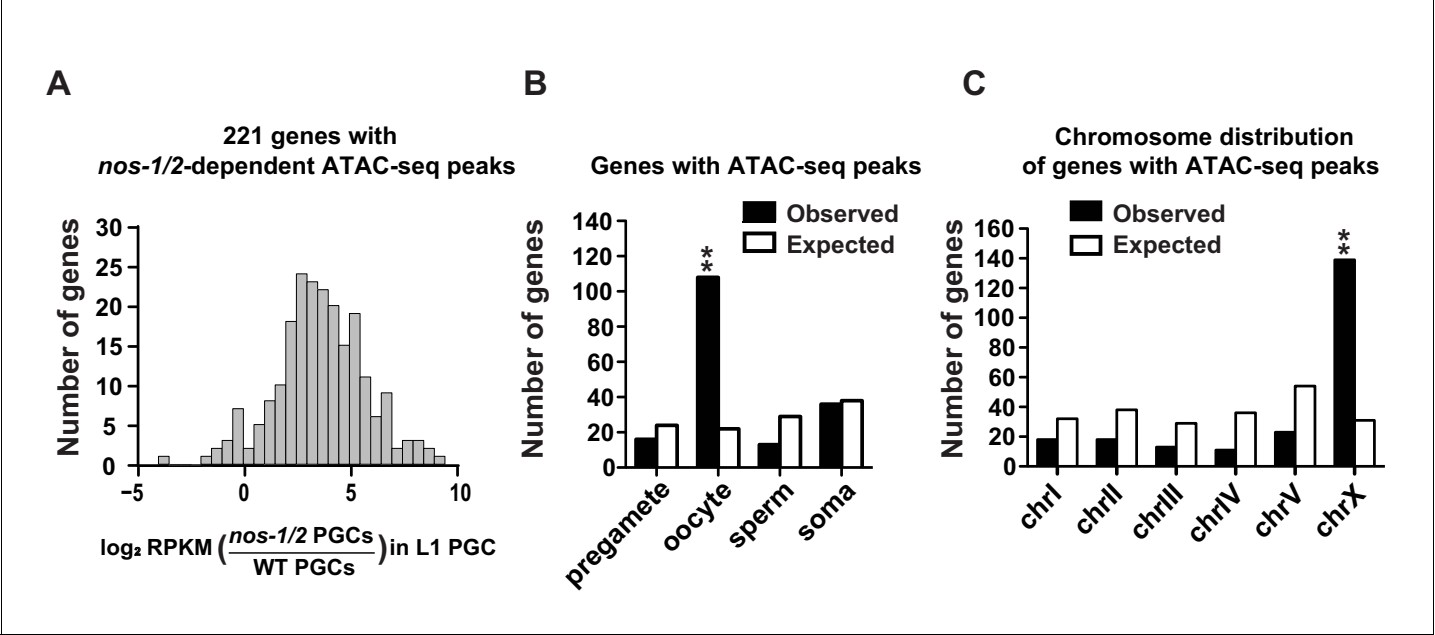

**Figure 3.** *nos-1nos-2* L1 PGCs activate the transcription of oocyte and X-linked genes. (**A**) Histogram showing the distribution of log2 fold change in gene expression between *nos-1(gv5)nos-2(RNAi)* (designated as *nos-1/2*) and wild-type L1 PGCs for 221 genes that acquired new ATAC-seq peaks in *nos-1(gv5)nos-2(RNAi)* PGCs (*Supplementary file 2*). Consistent with ATAC-seq peaks denoting open chromatin, most genes are expressed at higher levels in *nos-1(gv5)nos-2(RNAi)* PGCs compared to wild-type. (**B**) Bar graph showing expected and observed number of genes with *nos-1nos-2* - dependent ATAC-seq peaks in the different expression categories. Asterisks indicate significantly more genes than expected (hypergeometric test, p-value<0.001 [**]). (**C**) Bar graph showing the chromosomal distribution of genes with *nos-1nos-2* -dependent ATAC-seq peaks. Asterisks indicate significantly more genes than expected (hypergeometric test, p-value<0.001 [**]).
DOI: https://doi.org/10.7554/eLife.30201.010

The following figure supplements are available for figure 3:

**Figure supplement 1.** ATAC-seq reveals abnormal chromatin profile of *nos-1(gv5)nos-2(RNAi)* PGCs.
DOI: https://doi.org/10.7554/eLife.30201.011

**Figure supplement 2.** Analysis of reproducibility between ATAC-seq samples.
DOI: https://doi.org/10.7554/eLife.30201.012

not shown) for X-linked genes. Interestingly, the correlations were weaker for autosomal oocyte genes (R = 0.35, *Figure 4F*), which tended to be more overexpressed in *nos-1(gv5)nos-2(RNAi)* L1 PGCs. This finding is consistent with the notion that, while *nos-1nos-2* and *mes* PGCs share a defect in the silencing of X-linked loci, *nos-1nos-2* PGCs also have an additional defect in maternal mRNA turn over.

MES-2, 3, 4 and 6 proteins are maternally-inherited and are maintained in PGCs throughout embryogenesis (*Fong et al., 2002*; *Holdeman et al., 1998*; *Korf et al., 1998*; *Strome, 2005*). We observed no significant changes in *mes* transcripts in *nos-1(gv5)nos-2(RNAi)* PGCs compared to wild-type (*Supplementary file 4*). Direct examination of MES-2, MES-3 and MES-4 proteins confirmed that their expression patterns were unchanged in *nos-1(gv5)nos-2(ax3103)* or *nos-1(gv5)nos-2 (RNAi)* embryos (*Figure 4—figure supplement 2*). Together, these results suggest that *nos-1* and *nos-2* do not affect MES expression despite being required for MES-dependent silencing.

## Loss of *lin-15B, lin-35/RB* and *dpl-1/DP* suppresses *nos-1nos-2* sterility

MES-dependent silencing in somatic cells and adult germlines is antagonized by members of the synMuvB class of transcriptional regulators (*Petrella et al., 2011*; *Tabuchi et al., 2014*). To test whether synMuvB activity contributes to the *nos-1nos-2* PGC phenotype, we tested whether inactivation of synMuvB genes could reduce the sterility of *nos-1nos-2* animals using combinations of RNAi and mutants (*Figure 5—figure supplement 1*) and verified positives by analyzing the sterility of triple mutant combinations (*Figure 5*). We found that loss-of-function mutations in *lin-15B, lin-35* and

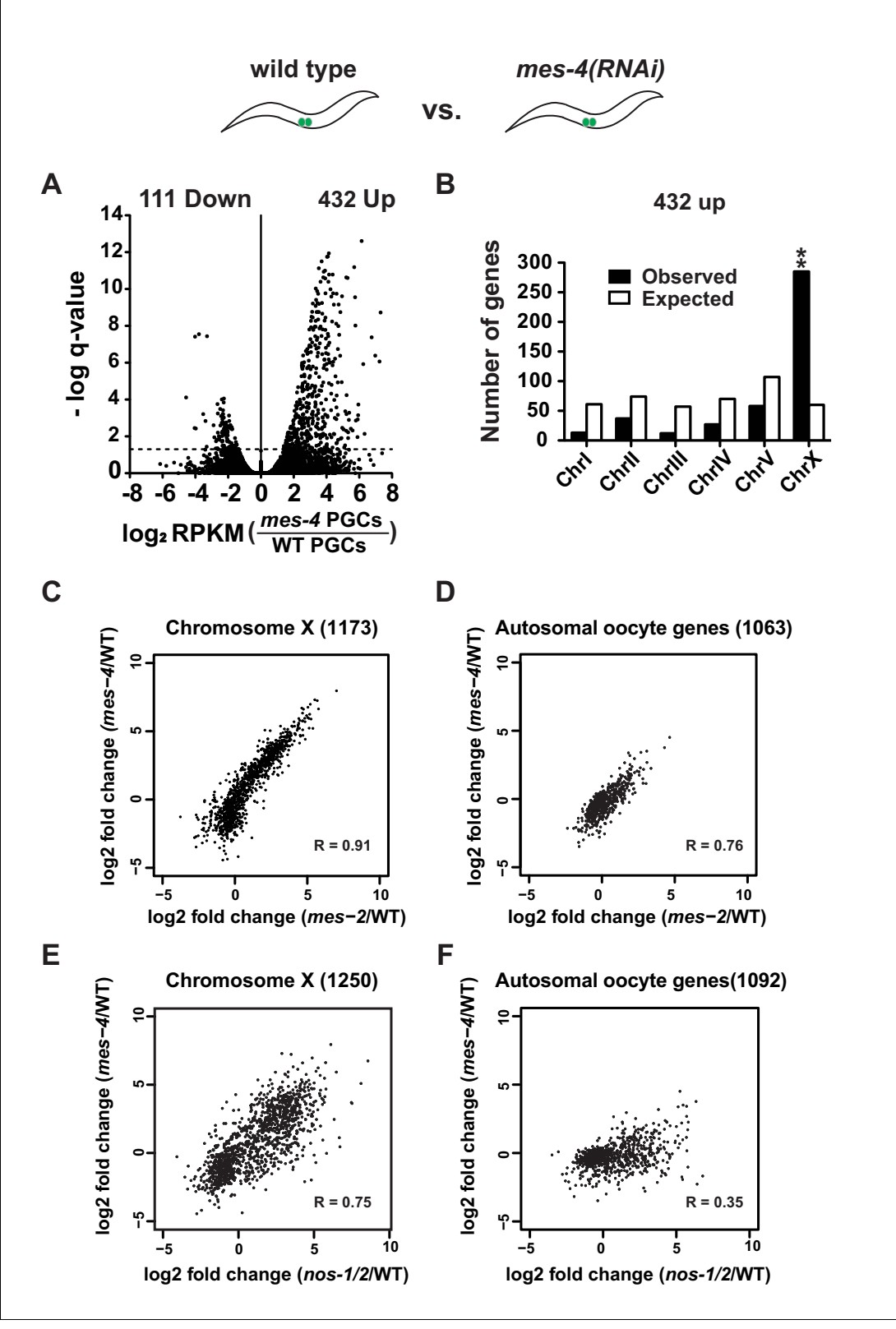

**Figure 4.** *mes-4* and *nos-1nos-2* PGCs exhibit a similar defect in X-chromosome silencing. (**A–B**) Transcriptome comparison between PGCs isolated from wild-type and *mes-4(RNAi)* L1 larvae. See **Figure 4—figure supplement 2** for comparison between wild-type and *mes-2(RNAi)*. (**A**) Volcano plot showing log2 fold-change in transcript abundance for each gene. The numbers of genes whose expression were up or downregulated in *mes-4(RNAi)* *Figure 4 continued on next page*

*Figure 4 continued*

PGCs are indicated. Dashed lines mark the significance cutoff of q = 0.05 above which genes were counted as misexpressed. (B) Bar graph showing chromosomal distribution of *mes-4(RNAi)* upregulated genes. Asterisks indicate significantly more genes than expected (hypergeometric test, p-value<0.001 [**]). (C–D) XY scatter plots showing correlation of the fold change in gene expression between *mes-2(RNAi)* (X-axis) and *mes-4(RNAi)*(Y-axis) PGCs compared to wild-type for X-linked genes and autosomal oocyte genes. Pearson correlation values are indicated. (E–F) XY scatter plots showing correlation of the fold change in gene expression between *nos-1(gv5) nos-2(RNAi)* (X-axis) and *mes-4(RNAi)* (Y-axis) PGCs compared to wild-type for X-linked genes and autosomal oocyte genes. Pearson correlation values are indicated.

DOI: https://doi.org/10.7554/eLife.30201.013

The following figure supplements are available for figure 4:

**Figure supplement 1.** *nos-1nos-2* PGCs share a defect in X chromosome silencing with *mes-4* PGCs.
DOI: https://doi.org/10.7554/eLife.30201.014

**Figure supplement 2.** MES proteins are expressed in *nos-1nos-2* embryonic PGCs.
DOI: https://doi.org/10.7554/eLife.30201.015

---

*dpl-1* reduced the sterility of *nos-1(gv5)nos-2(ax3103)* from >70% to<30%. (**Figure 5A**). The most dramatic reduction was seen with *lin-15B(n744)*, which reduced *nos-1(gv5)nos-2(ax3103)* sterility to 3.4% (**Figure 5**). *lin-15B* is a THAP domain DNA binding protein that has been implicated with the DRM class of transcriptional regulators, including *lin-35* and *dpl-1*, in the silencing of germline genes in somatic cells (**Araya et al., 2014**; **Petrella et al., 2011**; **Wu et al., 2012**). Other DRM components (*efl-1*, *lin-37*, *lin-9*, *lin-52*, *lin-54*), however, did not suppress *nos-1(gv5)nos-2(ax3103)* sterility (**Figure 5—figure supplement 1**).

Since PGCs lacking *mes* and *nos-1nos-2* shared the same defect in X chromosome silencing (**Figure 4E**, **Figure 4—figure supplements 1** and **2**), we tested whether loss of *lin-15B* could also suppress *mes-2* and *mes-4* maternal-effect sterility. Hermaphrodites derived from *mes-2(ok2480)* and *mes-4(ok2326)* mothers are 100% sterile (**Figure 5A** and **Figure 5—figure supplement 2**). We found that *lin-15B(n744)* suppressed *mes-2(ok2480)* and *mes-4(ok2326)* sterility weakly and only for one generation. Animals derived from *mes-2(ok2480);lin-15B(n744)* mothers were 83% sterile in the first generation and 98% sterile in the second generation and could not be maintained as a selfing population (**Figure 5A** and **Figure 5—figure supplement 2**). In contrast, *nos-1(gv5)nos-2(ax3103); lin-15B(n744)* triple mutant animals were almost fully fertile (96.6% fertile, **Figure 5A**) and could be maintained as a selfing population for >10 generations. The fertility of *nos-1(gv5)nos-2(ax3103);lin-15B(n744)* hermaphrodites was dependent on *mes* activity: inactivation by RNAi of *mes-2* or *mes-4* in *nos-1(gv5)nos-2(ax3103);lin-15B* resulted in 100% sterility (**Figure 5—figure supplement 3**). These genetic observations suggest that the sterility of *nos-1nos-2* mutants is due, at least in part to, inappropriate inhibition of MES function in PGCs by LIN-15B.

## Maternal LIN-15B is inherited by all embryonic blastomeres and downregulated specifically in PGCs

LIN-15B has been reported to be broadly expressed in somatic cells (**Sarov et al., 2012**), but its expression pattern during germline development was not known. We used a polyclonal antibody generated against LIN-15B protein (modencode project, personal communication with Dr. Susan Strome) to examine LIN-15B expression in the adult germline and in embryos. We confirmed the specificity of this antibody by staining *lin-15B(n744)* mutant, which showed no nuclear staining (**Figure 6—figure supplement 1**). We first detected LIN-15B expression in the germline in the L4 stage in nuclei near the end of the pachytene region where germ cells initiate oogenesis (**Figure 6A**). Nuclear LIN-15B was present in all oocytes and inherited by all embryonic blastomeres, including the germline P blastomeres (**Figure 6B** and **Figure 6—figure supplement 1**). LIN-15B remained present at high levels in all somatic nuclei throughout embryogenesis. In contrast, in the germ lineage, LIN-15B levels decreased sharply during the division of the germline founder cell P4 that generates the two PGCs (**Figure 6B** Left panels). LIN-15B expression remained at background levels in PGCs throughout embryogenesis.

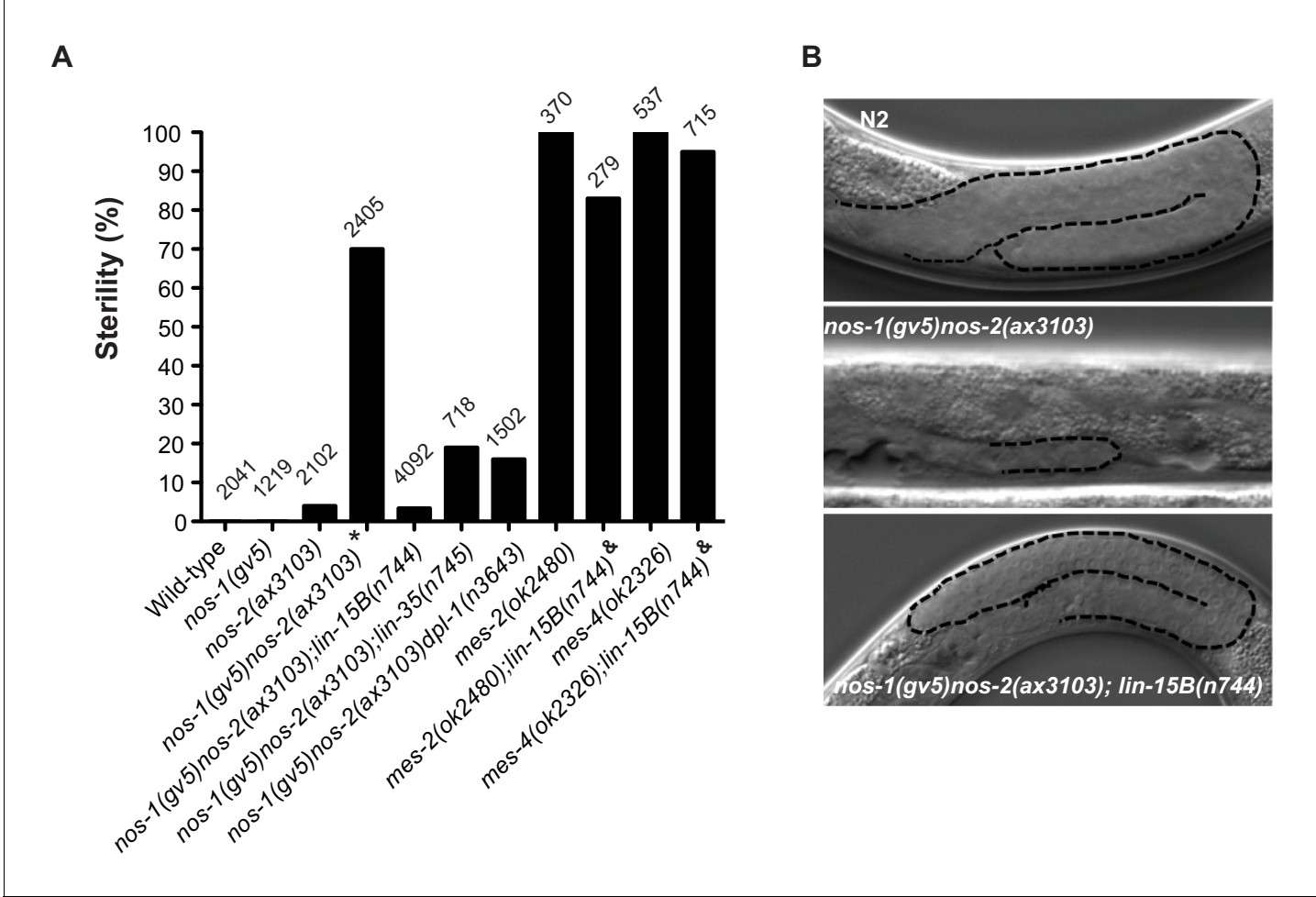

**Figure 5.** Suppression of *nos-1nos-2* sterility by *lin-15b* and *synMuvB* mutants. (**A**) Bar graph showing the % sterility at 20°C among progenies of hermaphrodites with the listed genotypes. *lin-15B(n744)* and *lin-35(n745)* are null alleles(*Ferguson and Horvitz, 1989*; *Lu and Horvitz, 1998*; *Petrella et al., 2011*). *dpl-1(n3643)* is a loss of function allele (*Ceol and Horvitz, 2001*) *mes-2(ok2480)* and *mes-4(ok2326)* are deletion alleles that causes 100% maternal-effect sterility (*C. elegans Deletion Mutant Consortium et al., 2012*). Number of progenies scored is written above indicated genotypes. * *nos-1(gv5)nos-2(ax3103)* hermaphrodites produce 70% sterile progenies at 20°C and 96% sterile progeny at 25°C with severely atrophied germlines (*Figure 5B*). *nos-1(gv5)nos-2(ax3103);lin-15B(n744)* hermaphrodites produce 96.6% fertile progenies at 20°C, and arrest as larvae at 26°C as is true of *lin-15B(n744)* animals. & *mes-2(ok2480);lin-15B(n744)* and *mes-4(ok2326);lin-15B(n744)* hermaphrodites cannot be maintained as selfing populations. (**B**) Nomarski images of germlines (stippled) in L4 hermaphrodites of the indicated genotypes. Worms were staged according to vulva morphology. Note the atrophied germline in *nos-1(gv5)nos-2(ax3103)* that is rescued to wild-type size in *nos-1(gv5)nos-2(ax3103);lin-15B(n744)*.
DOI: https://doi.org/10.7554/eLife.30201.016

The following figure supplements are available for figure 5:

**Figure supplement 1.** Suppression of *nos-1nos-2* sterility by *synMuvB* mutants.
DOI: https://doi.org/10.7554/eLife.30201.017

**Figure supplement 2.** Loss of *lin-15B* does not rescue *mes* maternal effect sterility.
DOI: https://doi.org/10.7554/eLife.30201.018

**Figure supplement 3.** Suppression of *nos-1nos-2* sterility by loss of *lin-15B* depends on *mes* activity.
DOI: https://doi.org/10.7554/eLife.30201.019

## Downregulation of maternal LIN-15B in PGCs requires *nos-1 nos-2* activity

*lin-15B* transcripts were modestly elevated in *nos-1(gv5)nos-2(RNAi)* EMB PGCs compared to wild-type EMB PGCs, suggesting that *lin-15B* may be one of the maternal RNAs that requires Nanos activity for rapid turnover in PGCs (*Supplementary file 4*). During the transition from EMB to L1

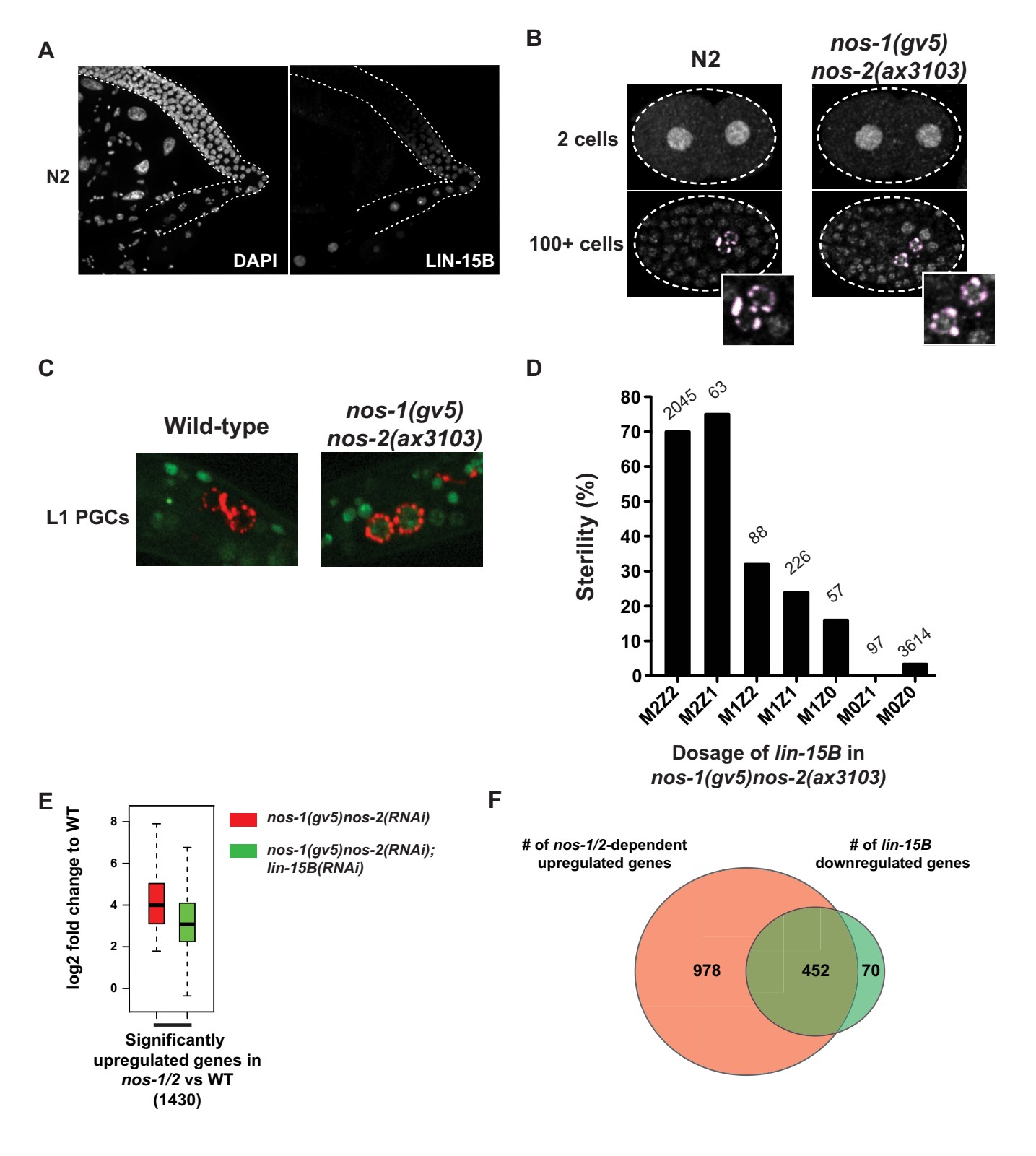

**Figure 6.** LIN-15B is inherited maternally and is downregulated in PGCs in a *nos-1nos-2* dependent manner. (**A**) Photomicrograph of a dissected wild-type gonad stained with anti-LIN-15B antibody and DAPI for DNA. LIN-15B protein is detected at the end of the pachytene region and in all oocyte nuclei. (**B**) Photomicrographs of fixed wild-type and *nos-1(gv5)nos-2(ax3103)* embryos stained with α-LIN-15B and K76 (α-PGL-1, pink) antibodies. The α-LIN-15B polyclonal serum cross-reacts with perinuclear germ granules (pink color, see Materials and methods). 45/60 PGCs were positive for LIN-15B in

*Figure 6 continued on next page*

*Figure 6 continued*

*nos-1(gv5)nos-2(ax3103)* embryos compared to 0/34 in wild-type. (C) Photomicrographs of newly hatched gonads from wild-type and *nos-1(gv5)nos-2 (ax3103)* L1 larvae with a paternal copy of the *lin-15B* transcriptional reporter (green). 12/16 *nos-1(gv5)nos-2(ax3103)* PGC doublets were positive for GFP compared to 0/28 in wild-type. See *Figure 6—figure supplement 1B and C* for description of the *lin-15B* transcriptional reporter. (D) Bar graph showing the sterility of *nos-1(gv5)nos-2(ax3103)* hermaphrodites with different dosages of maternal (M) and zygotic (Z) *lin-15B* at 20℃. M2Z2 denotes hermaphrodites with two doses of wild-type maternal LIN-15B and two doses of wild-type zygotic LIN-15B. Mating schemes are shown in *Figure 6— figure supplement 2*. Number of hermaphrodites scored is written above each genotype. (E) Box and Whisker plot showing log2 fold change compared to wild-type of 1430 genes that are upregulated in *nos-1(gv5)nos-2(RNAi)* (designated as *nos-1/2*) L1 PGCs. Each box extends from the 25th to the 75th percentile, with the median indicated by the horizontal line; whiskers extend from the 2.5th to the 97.5th percentiles. The upregulation is reduced in *nos-1(gv5)nos-2(RNAi);lin-15B(RNAi)* PGCs. See *Figure 6—figure supplement 3* for additional comparisons. (F) Venn diagram showing overlap between 1430 genes upregulated in *nos-1(gv5)nos-2(RNAi)* (designated as *nos-1/2*) compared to wild-type L1 PGCs (red) and downregulated genes in *nos-1(gv5)nos-2(RNAi);lin-15B(RNAi)* compared to *nos-1(gv5)nos-2(RNAi)* L1 PGCs (522 genes, green).

DOI: https://doi.org/10.7554/eLife.30201.020

The following figure supplements are available for figure 6:

**Figure supplement 1.** Expression of *lin-15B* begins in late pachytene germ cells.

DOI: https://doi.org/10.7554/eLife.30201.021

**Figure supplement 2.** Assay for maternal and zygotic contribution of *lin-15B* in *nos-1nos-2* sterility.

DOI: https://doi.org/10.7554/eLife.30201.022

**Figure supplement 3.** Loss of *lin-15B* activity mitigates gene expression changes in *nos-1nos-2* PGCs.

DOI: https://doi.org/10.7554/eLife.30201.023

PGCs, *lin-15B* transcript levels rose by ~2 fold in *nos-1(gv5)nos-2(RNAi)* embryos but not in wild-type embryos, suggesting that the *lin-15B* locus is also inappropriately transcribed in *nos-1(gv5)nos-2 (RNAi)* L1 PGCs. Unfortunately, we were not able to confirm these RNA-seq observations by in situ hybridization due to the low abundance of *lin-15B* RNA and its presence in all somatic cells.

To determine whether LIN-15B protein expression is under the control of Nanos activity, we stained embryos derived from *nos-1(gv5)nos-2(ax3103)* hermaphrodites with the anti-LIN-15B antibody. We found that, in contrast to wild-type, *nos-1(gv5)nos-2(ax3103)* embryos maintained high LIN-15B levels in embryonic PGCs (*Figure 6B* Right panels). *nos-1(gv5)nos-2(ax3103)* embryos could misregulate LIN-15B by delaying the turnover of maternal LIN-15B or by activating premature zygotic transcription of the *lin-15B* locus. To distinguish between these possibilities, we created a *lin-15B* transcriptional reporter by inserting a GFP::H2B fusion at the 5' end of *lin-15B* locus in an operon configuration to preserve endogenous *lin-15B* expression (*Figure 6—figure supplement 1* and *Supplementary file 6*). We crossed *nos-1(gv5)* males carrying the *lin-15B* transcriptional reporter to wild-type or *nos-1(gv5)nos-2(ax3103)* hermaphrodites and examined crossed progenies for GFP expression. In both cases, we observed strong GFP expression in somatic cells, but no expression in PGCs during embryogenesis (data not shown), indicating that zygotic expression of LIN-15B in PGCs is not activated in embryogenesis in either wild-type or *nos-1(gv5)nos-2(ax3103)* embryos. In wild-type, we first observed zygotic expression of the *lin-15B* transcriptional reporter in the germline of L4 stage animals (*Figure 6—figure supplement 1*), in germ cells that have initiated oogenesis. In contrast, in animals derived from *nos-1(gv5)nos-2(ax3103)* mothers, zygotic expression of the *lin-15B* transcriptional reporter could be detected as early as the L1 stage in PGCs and their descendants (*Figure 6C*). This expression was maintained until the L2 stage when *nos-1nos-2* PGC descendants undergo cell death. We conclude that *nos-1nos-2* activity is required both to promote the turnover of maternal LIN-15B in PGCs during embryonic development and to prevent premature zygotic transcription of the *lin-15B* locus in PGCs in the L1 stage.

## Maternal *lin-15B* is responsible for *nos-1 nos-2* sterility

To determine whether misregulation of maternal or zygotic *lin-15B* is responsible for *nos-1nos-2* sterility, we compared the sterility of *nos-1(gv5)nos-2(ax3103)* animals that lack either maternal or zygotic *lin-15B* (*Figure 6D* and *Figure 6—figure supplement 2*). We found that loss of maternal *lin-15B* was sufficient to fully suppress *nos-1(gv5)nos-2(ax3103)* sterility, even in the presence of one zygotic copy of *lin-15B* (*Figure 6D*). The penetrance of the suppression was dependent on the dosage of maternal *lin-15B*. *nos-1(gv5)nos-2(ax3103)* animals with only one copy of maternal *lin-15B* were only 32% sterile compared to 70% sterility for animals with two copies of maternal *lin-15B* and

0% with animals with zero copies of maternal *lin-15B* (*Figure 6D*). Interestingly, animals with only one copy of maternal LIN-15B appeared sensitive to the zygotic dosage of *lin-15B* (*Figure 6D*, compare the sterility M1Z2, M1Z1 and M1Z0). We conclude that maternal *lin-15B* is primarily responsible for the sterility of *nos-1nos-2* animals, although zygotic Lin-15B activity may also contribute.

### Loss of *lin-15B* activity mitigates gene expression changes in *nos-1 nos-2* PGCs

LIN-15B is a transcription factor with many targets in somatic cells but no known function in the germline (*Niu et al., 2011*). To determine the effect of ectopic LIN-15B on the transcriptome of *nos-1(gv5)nos-2(RNAi)* PGCs, we profiled *nos-1(gv5)nos-2(RNAi);lin-15B(RNAi)* PGCs and compared the log2 fold change of transcripts in *nos-1(gv5)nos-2(RNAi);lin-15B(RNAi)* PGCs and *nos-1(gv5)nos-2(RNAi)* PGCs to wild-type. We found that loss of *lin-15B* reduced gene misexpression in *nos-1(gv5)nos-2(RNAi)* PGCs (*Figure 6E*). Of the 1430 upregulated genes in *nos-1(gv5)nos-2(RNAi)* PGCs, 31% (452) had significantly lower expression levels in *nos-1(gv5)nos-2(RNAi);lin-15B(RNAi)* PGCs (*Figure 6F*). Both upregulated and down-regulated gene categories were at least partially rescued, as well as X-linked and oocyte genes (*Figure 6—figure supplement 3*). These data indicate that ectopic *lin-15B* activity is responsible for a significant number of misexpressed genes in *nos-1(gv5)nos-2(RNAi)* PGCs.

To determine whether loss of *lin-15B* also rescues the defect in maternal RNA turnover in *nos-1nos-2* PGCs, we performed in situ hybridization on *nos-1(gv5)nos-2(ax3103);lin-15B(n744)* embryos. We found that the turnover of *mex-5* and C01G8.1 mRNAs was still delayed in these embryos, as is observed in *nos-1(gv5)nos-2(ax3103)* (*Figure 2—figure supplement 2*). We conclude that loss of *lin-15B* does not rescue the delay in maternal mRNA turnover observed in *nos-1(gv5)nos-2(ax3103)* PGCs.

By comparing the lists of upregulated genes in *nos-1(gv5)nos-2(RNAi)* and *mes-4(RNAi)* PGCs and of down-regulated genes in *nos-1(gv5)nos-2(RNAi);lin-15B(RNAi)* PGCs (*Supplementary file 5*), we identified 88 shared genes, including 70 X-linked genes. Among these is *utx-1*, an histone demethylase specific for the H3K27me3 mark generated by *mes-2* (*Agger et al., 2007*; *Seelk et al., 2016*). Like other X-linked genes, *utx-1* transcripts are rare in wild-type PGCs (FPKM <0.2) (*Supplementary file 4*) and are overexpressed 9.1-fold in *mes-2(RNAi)* PGCs. In *nos-1(gv5) nos-2 (RNAi)* L1 PGCs, the *utx-1* locus acquires a new ATAC-seq peak (*Figure 3—figure supplement 1*) and *utx-1* transcripts are overexpressed 160-fold. This overexpression was reduced significantly by 2.4-fold in *nos-1(gv5)nos-2(RNAi);lin-15B(RNAi)* PGCs. These observations suggest that *utx-1* may function downstream or in parallel to *lin-15B* to further antagonizes MES activity as X-linked genes become desilenced. If so, loss of *utx-1* should alleviate *nos-1nos-2* sterility. Consistent with this prediction, we found that reduction of *utx-1* activity by RNAi partially suppressed *nos-1(gv5)nos-2(ax3103)* sterility (*Figure 5—figure supplement 1*). Suppression by *utx-1* was not as extensive as that observed with *lin-15B*, suggesting that *utx-1* is not the only gene activated in *nos-1(gv5)nos-2(ax3103)* PGCs that leads to sterility. These results suggest that activation of *utx-1* may participate in a regulatory loop downstream of maternal LIN-15B that further weakens *mes* activity in *nos-1nos-2* animals.

## Discussion

In this study, we have examined the transcriptome of PGCs lacking Nanos function in *C. elegans*. We have found that PGCs lacking Nanos overexpress 100 s of transcripts normally expressed in oocytes. Our observations indicate that this defect is due to both delayed turn-over of maternal mRNAs and inappropriate transcription of genes normally silent in PGCs, including oocyte genes. We identify LIN-15B as a critical maternally-inherited factor that must be turned-over by Nanos in PGCs to prevent inappropriate transcription. We propose that clearing of maternal LIN-15B by Nanos allows the PRC2/MES-4 network of chromatin modifiers to reprogram PGCs away from an inherited (maternal) oocyte program to a pregametic germ cell program.

## Nanos activity is required for the timely turnover of maternal mRNAs in PGCs

During oogenesis, oocytes stockpile mRNAs and proteins in preparation for embryogenesis. These include mRNAs and proteins with housekeeping functions as well as factors required to specify embryonic cell fates (somatic and germline). During embryogenesis, maternal products are eventually turned over to make way for zygotic factors (maternal-to-zygotic transition). Our findings suggest that Nanos facilitates this transition in PGCs by accelerating the turnover of maternal mRNAs. Nanos family members are thought to silence mRNAs by interacting with the sequence-specific RNA-binding protein Pumilio and with the CCR4-NOT deadenylase complex, which interferes with translation and can also destabilize RNAs. (*Lai et al., 2012a*; *Suzuki et al., 2012*; *Swartz et al., 2014*; *Wharton et al., 1998*). In the *C. elegans* genome, there are eight genes related to *Drosophila pumilio*. Depletion of five of these (*fbf-1, fbf-2, puf-6, puf-7 and puf-8*) phenocopies the *nos-1nos-2* PGC phenotypes, including failure to incorporate in the somatic gonad, premature proliferation, and eventually cell death (*Subramaniam and Seydoux, 1999*). These observations suggest that NOS-1 and NOS-2 function with Pumilio-like proteins to repress the translation of certain maternal RNAs. Paradoxically, in sea urchins, Nanos silences the mRNA coding for the CNOT6 deadenylase, which indirectly stabilizes other maternal mRNAs (*Swartz et al., 2014*). In that system, Nanos was also found to silence eEF1A expression, leading to a transient period of translational quiescence in PGCs (*Oulhen et al., 2017*). In combination, these effects could promote the turnover of maternal mRNAs and proteins that promote somatic development (e.g. LIN-15B) while preserving germline mRNAs (e.g. *mes*) whose translation could be reactivated at a later time. In *C. elegans*, the redundant *nanos* homologs *nos-1* and *nos-2* are expressed sequentially in PGCs during the maternal-to-zygotic transition and may have distinct targets. Genetic analyses already have suggested that *nos-1* and *nos-2* have both shared and unique functions (*Kapelle and Reinke, 2011*; *Mainpal et al., 2015*). It will be important to determine whether *nos-1* and *nos-2* both target *lin-15B*, and whether they do so directly, by silencing *lin-15B* mRNA translation, or indirectly, by silencing other factors required for LIN-15B protein translation and/or stability.

## In the absence of Nanos, maternal LIN-15B interferes with MES-dependent reprogramming of PGC chromatin

Two lines of evidence indicate that LIN-15B is responsible for much of the abnormal gene expression observed in *nos-1nos-2* PGCs by the first larval stage. First, elimination of maternal LIN-15B restores fertility to *nos-1nos-2* mutants and lessens the upregulation of many misregulated genes (*Figure 6*). Second, LIN-15B is a known genetic antagonist of MES function in somatic cells (*Petrella et al., 2011*; *Wang et al., 2005*), and PGCs lacking *mes* activity upregulate many of the same genes upregulated in *nos-1nos-2* PGCs. The strongest correlation is seen for genes on the X chromosome (*Figure 4E*, *Figure 4—figure supplement 1*), a well-documented focus of MES-dependent silencing (*Bender et al., 2006*; *Garvin et al., 1998*; *Gaydos et al., 2012*). Together these findings indicate that failure to downregulate maternal LIN-15B interferes with MES-dependent reprogramming of PGC chromatin and is the primary cause of PGC death in Nanos mutants.

The *lin-15B* locus is on the X chromosome and is ectopically transcribed in *nos-1nos-2* PGCs at hatching. These observations raise the possibility that maternal LIN-15B potentiates zygotic *lin-15B* expression as MES-dependent silencing of the X-chromosome becomes compromised. How LIN-15B opposes MES activity is not known, but another X-linked gene, *utx-1,* may oppose MES activity directly. *utx-1* encodes a de-methylase that removes the silencing mark deposited by the PRC2 complex. Upregulation of *utx-1* was shown recently to promote reprogramming of adult germline stem cells into neurons (*Seelk et al., 2016*). In *nos-1nos-2* PGCs, *utx-1* is upregulated in a *lin-15B*-dependent manner, and RNAi of *utx-1* partially suppresses *nos-1nos-2* sterility (*Figure 5—figure supplement 1*). Suppression by loss of *utx-1* is weaker than that observed when inactivating *lin-15B*, suggesting that *utx-1* is not the only *lin-15B* target that opposes PRC2. Loss of two other synMuvB genes, *lin-35/Rb* and *dpl-1,* also suppresses *nos-1nos-2* sterility (*Figure 5A*), albeit again less stringently than loss of *lin-15B*. It will be interesting to determine whether these genes function with, or in parallel to, LIN-15B to oppose PRC2 activity in PGCs.

Recently, *nos-2* was shown to function redundantly with *xnd-1* to repress histone active marks in PGCs (*Mainpal et al., 2015*). XND-1 is a chromatin-associated protein expressed in PGCs

throughout embryogenesis. An exciting possibility is that XND-1 directly activates MES-dependent remodeling in PGCs. In that context, Nanos could promote germ cell fate simply by eliminating any maternal factors that would interfere with that remodeling. Because PGCs derive from embryonic blastomeres that also give rise to somatic lineages, they inherit many transcripts with somatic functions. In addition to LIN-15B, we have found that Nanos accelerates the turn-over of several maternal mRNAs coding for transcription factors that function in somatic embryonic lineages, including *pha-4*, *hlh-1* and *tbx-2* (*Supplementary file 4*). We speculate that perdurance of these somatic transcription factors contributes to the complex transcriptional profile of *nos-1nos-2* PGCs. The primary function of Nanos may be, therefore, to clear the PGCs of any mRNAs that promote somatic development. This interpretation is consistent with previous studies in Drosophila and Xenopus that reported the expression of somatic transcripts in PGCs lacking Nanos (*Deshpande et al., 1999b*; *Hayashi et al., 2004*; *Kadyrova et al., 2007*; *Lai et al., 2012b*; *Oulhen et al., 2017*; *Swartz et al., 2014*).

### Nanos functions in an ancient regulatory switch that controls somatic and germline fates throughout development?

Our genetic findings indicate that, in PGCs, Nanos opposes LIN-15B and DRM transcription factors. Studies in Drosophila and mammals have reported that, in somatic cells, the reverse is true: DRM transcription factors silence Nanos. Loss of the DRM subunit *lethal (3) malignant brain tumor [l(3) mbt]* leads to tumorous growth in Drosophila imaginal disks and ectopic expression of germline genes, including *nanos* (*Janic et al., 2010*). Similarly, loss of the retinoblastoma transcription factor (Rb) leads to activation of *nanos* transcription in mammalian tissue culture cells and in *Drosophila* wings, which in turn is thought to repress the translation of Rb targets (*Miles and Dyson, 2014*; *Miles et al., 2014*). A complex regulatory feedback loop has also been reported between the LSD1 demethylase and the Nanos partner Pumilio in Drosophila and human bladder carcinoma cells (*Miles et al., 2015*). Taken together, these observations suggest that Nanos functions in an ancient transcriptional/post-transcriptional regulatory switch that controls gene expression during development. Key questions for the future will be to understand how the switch is activated in the embryonic germline to favor germ cell development (what turns on Nanos expression in PGCs?), how the switch is flipped back during oogenesis to favor somatic development (what activates LIN-15B expression in oocytes and embryos?), and how the switch becomes deregulated in malignancies.

## Materials and methods

**Key resources table**

| Reagent type (species) or resource | Designation | Source or reference | Identifiers | Additional information |
|---|---|---|---|---|
| strain, strain background (C. elegans) | JH1270 | *Subramaniam and Seydoux (1999)* | RRID: WB-STRAIN:JH1270 | *nos-1(gv5)* |
| strain, strain background (C. elegans) | JH3103 | This study | | *nos-1(gv5); lin-15A(n767)* |
| strain, strain background (C. elegans) | TH206 | (http://www.moden code.org). | RRID: WB-STRAIN:TH206 | *unc-119(ed3) III; ddEx16* |
| strain, strain background (C. elegans) | JH3099 | This study | | *unc-119(ed3) III; ddEx16 out cross x2* |
| strain, strain background (C. elegans) | MT1806 | CGC | RRID: WB-STRAIN:MT1806 | *lin-15A(n767)* |
| strain, strain background (C. elegans) | PFR40 | CGC | RRID: WB-STRAIN:PFR40 | *hpl-2(tm1489)* |
| strain, strain background (C. elegans) | MT2495 | CGC | RRID: WB-STRAIN:MT2495 | *lin-15B(n744)* |
| strain, strain background (C. elegans) | MT10430 | CGC | RRID: WB-STRAIN:MT10430 | *lin-35(n745)* |
| strain, strain background (C. elegans) | MT11147 | CGC | RRID: WB-STRAIN:MT10430 | *dpl-1(n3643)* |

*Continued on next page*

*Continued*

| Reagent type (species) or resource | Designation | Source or reference | Identifiers | Additional information |
|---|---|---|---|---|
| strain, strain background (C. elegans) | JH3109 | This study | | *nos-1(gv5); hpl-2(tm1489)* |
| strain, strain background (C. elegans) | JH3119 | This study | | *nos-1(gv5); lin-35(n745)* |
| strain, strain background (C. elegans) | JH3121 | This study | | *nos-1(gv5); lin-15B(n744)* |
| strain, strain background (C. elegans) | JH3141 | This study | | *nos-1(gv5) dpl-1(n3643)* |
| strain, strain background (C. elegans) | JH3357 | This study | | *nos-2(ax3103)* |
| strain, strain background (C. elegans) | JH3367 | This study | | *nos-1(gv5) nos-2(ax3103)/MnC1* |
| strain, strain background (C. elegans) | JH3401 | This study | | *nos-1(gv5); nos-2(ax3103); lin-15B(n744)* |
| strain, strain background (C. elegans) | JH3428 | This study | | *mes-2(ax2509 [mes-2::GFP]); tagRFP::glh-1* |
| strain, strain background (C. elegans) | JH3436 | This study | | *tagRFP::glh-1; nos-1(gv5); lin-15B(ax3104)* |
| strain, strain background (C. elegans) | JH3484 | This study | | *mes-3(ax3105 [mes-3::OLLAS])* |
| strain, strain background (C. elegans) | JH3486 | This study | | *mes-3(ax3105 [mes-3::OLLAS]); nos-1(gv5) nos-2 (ax3103)/MnC1* |
| strain, strain background (C. elegans) | JH3203 | CGC | RRID: WB-STRAIN:JH3203 | *mes-2(ax2059 [mes-2::GFP])* |
| strain, strain background (C. elegans) | JH3510 | This study | | *mes-2(ax2509 [mes-2::GFP]); tagRFP::glh-1; nos-1(gv5) nos-2(ax3103)/MnC1* |
| strain, strain background (C. elegans) | JH3513 | This study | | *gtbp-1(axIs3105[gtbp-1 prom:: GFP-H2B::lin-15B 3'utr]); tagRFP::glh-1; nos-1(gv5) nos-2(ax3103)/MnC1* |
| strain, strain background (C. elegans) | JH3531 | This study | | *dpl-1(n3643) nos-1(gv5)nos-2(ax3103)* |
| strain, strain background (C. elegans) | JH3538 | This study | | *lin-35(n745); nos-1(gv5) nos-2(ax3103)* |
| strain, strain background (C. elegans) | VC2409 | CGC | RRID: WB-STRAIN:VC2409 | *mes-2(ok2480)/mT1 II; +/mT1 [dpy-10(e128)] III* |
| strain, strain background (C. elegans) | VC1874 | CGC | RRID: WB-STRAIN:VC1874 | *mes-4(ok2326) V/nT1[qIs51] (IV;V)* |
| strain, strain background (C. elegans) | JH3357 | This study | | *nos-2(ax3103).* Deletion of *nos-2* ORF. See ***Supplementary file 6*** for description. |

*Continued on next page*

*Continued*

| Reagent type (species) or resource | Designation | Source or reference | Identifiers | Additional information |
|---|---|---|---|---|
| strain, strain background (C. elegans) | JH3436 | This study | | *lin-15B(ax3104). lin-15B prom::GFP-H2B::tbb-2 3'UTR.* See *Supplementary file 6* for description. |
| strain, strain background (C. elegans) | JH3484 | This study | | *mes-3(ax3105). mes-3::OLLAS.* See *Supplementary file 6* for description. |
| antibody | K76 | DSHB,PMID:28787592 | RRID:AB_531836 | (1:15) |
| antibody | Anti-FLAG M2 | Sigma-Aldrich Cat# F3165 | RRID:AB_259529 | (1:200) |
| antibody | Donkey-anti-mouse IgM 647 | Jackson Immuno Research Labs | RRID:AB_2340861 | (1:400) |
| antibody | Goat anti-Rabit IgG (H + L) 568 | Molecular probes cat# A-11011 | RRID:AB_143157 | (1:400) |
| antibody | Anti-OLLAS-L2 | Novus cat# NBP1-06713 | RRID:AB_1625979 | (1:200) |
| antibody | anti-LIN-15B | other | | gift from Dr. Susan Strome, SDQ3183 1:40,000 |
| antibody | anti-MES-4 | other | | gift from Dr. Susan Strome. (1:400) |
| antibody | anti-OLLAS | other | | gift from Dr. Jeremy Nathans (1:80) |
| sequence-based reagent | oCYL584 | This study | | GAUCUUCUAGAAAGAAUCUU; crRNA cut at 3' end of *nos-2* |
| sequence-based reagent | oCYL669 | This study | | AGAGUCGAAGUCGGUUCACU; crRNA cut at 5' end of *nos-2* |
| sequence-based reagent | oCYL823 | This study | | GCACUGCUACUGCUGGAAGU; crRNA cut at 5' end of *lin-15B* |
| sequence-based reagent | oCYL957 | This study | | GGGAUAAUCTAAUUAGAAGA; crRNA cut at 3' end of *mes-3* |
| sequence-based reagent | AP691 | *Paix et al. (2015)* | | GGCCTTAACCCAGAATAAGA; crRNA cut at 5' end of *gtbp-1* |
| sequence-based reagent | AP728 | *Paix et al. (2015)* | | CACGAGGTGGTATGCGCAG; crRNA cut at 3' end of *gtbp-1* |
| sequence-based reagent | oCYL251 | This study | | TGGAAAGTTGAGTGTGAGCA; Forward K08A8.1 RT-PCR primer |
| sequence-based reagent | oCYL252 | This study | | GGAGAATGTTTGATGGCTTCAC; Reverse K08A8.1 RT-PCR primer |
| sequence-based reagent | oCYL259 | This study | | CCTGAGAAGAAGCTGCAAATG; Forward W02A11.8 RT-PCR primer |
| sequence-based reagent | oCYL260 | This study | | TTTATGTCCTTTGGCAAAACGG; Reverse W02A11.8 RT-PCR primer |
| sequence-based reagent | oCYL304 | This study | | CTGCTATTGTGAAGTCTCCTG; Forward B0416.5 RT-PCR primer |
| sequence-based reagent | oCYL305 | This study | | CCATTTGTGGCTTACTAGCG; Reverse B0416.5 RT-PCR primer |
| sequence-based reagent | oCYL308 | This study | | TGTCAGTTTGTGATGTGCTG; Forward C35C5.3 RT-PCR primer |
| sequence-based reagent | oCYL309 | This study | | GCTTCAAAATCGTCCTTTTCATG; Reverse C35C5.3 RT-PCR primer |

*Continued on next page*

*Continued*

| Reagent type (species) or resource | Designation | Source or reference | Identifiers | Additional information |
|---|---|---|---|---|
| sequence-based reagent | oCYL738 | This study | | ACTGGACGATTTCAACGGAG; Forward *lin-15B* RT-PCR primer |
| sequence-based reagent | CYL739 | This study | | ACATACTGCACAGCGACG; Reverse *lin-15B* RT-PCR primer |
| sequence-based reagent | oCYL994 | This study | | AGTCGGTATTTTGAATGCGG; Forward *lsd-1* RT-PCR primer |
| sequence-based reagent | oCYL995 | This study | | CGTTTCCGAGTGATCTGATTG; Reverse *lsd-1* RT-PCR primer |
| sequence-based reagent | oCYL998 | This study | | AATCCGTTTGACTATGAGTGG; Forward W05H9.2 RT-PCR primer |
| sequence-based reagent | oCYL999 | This study | | TCGTTTAGAAGCTACAATGACAG; Reverse W05H9.2 RT-PCR primer |
| sequence-based reagent | oCYL1006 | This study | | GAAGTTACCCGTCGCAAG; Forward F28H6.4 RT-PCR primer |
| sequence-based reagent | oCYL1007 | This study | | GCCACTGTTTTGTAATCCCG; Reverse F28H6.4 RT-PCR primer |
| sequence-based reagent | oCYL1010 | This study | | ACTTTGCGATAAACTCCCTTC; Forward *tag-299* RT-PCR primer |
| sequence-based reagent | oCYL1011 | This study | | GCTTGCAGACACGAAGATAAG; Reverse *tag-299* RT-PCR primer |
| sequence-based reagent | oCYL1020 | This study | | CGAATGCGGACATCTTAATCC; Forward *lnp-1* RT-PCR primer |
| sequence-based reagent | oCYL1021 | This study | | GTTGACGGCTTCTGATTCTC; Reverse *lnp-1* RT-PCR primer |
| sequence-based reagent | oCYL1044 | This study | | TGGTTATGTGCAACACTTGG; Forward *sygl-1* RT-PCR primer |
| sequence-based reagent | oCYL1045 primer | This study | | TCTCGCTACGATCCTTCTTC; Reverse *sygl-1* RT-PCR |
| sequence-based reagent | oCYL438 | This study | | CAGCTCGAAACCTGAAAATTGT; Forward PCR primer for *nos-2* locus. 179 bp upstream of *nos-2* ATG. |
| sequence-based reagent | oCYL734 | This study | | GCCATCACCTATGCGATTTG; Reverse PCR primer for *nos-2* locus. 468 bp after *nos-2* STOP. |
| sequence-based reagent | oCYL735 | This study | | GTTGTGGCGGAAAGGAATAC; Reverse PCR primer for *nos-2* locus, 154 bp after *nos-2* ATG. |
| sequence-based reagent | oCYL43 | This study | | ATGTTGATTTTCAGGACTTCTC; Forward PCR primer for *nos-1* locus. seq from + 1- + 22 |
| sequence-based reagent | oCYL45 | This study | | ACGAAGCATCACCTGGAG; Forward PCR primer for *nos-1* locus. seq from + 901–918 (ORF seq). |
| sequence-based reagent | oCYL407 | This study | | CGTTGAAACTTTGAAGAAAGACATC; Forward PCR primer for *nos-1* locus. seq from + 901–918 (ORF seq). |
| sequence-based reagent | oCYL361 | This study | | GATGATTGTTGGAGAGGACG; Reverse PCR primer for *lin-15B* locus. Pair with oCYL363 to generated a PCR fragment contains n744 mutation. |

*Continued on next page*

*Continued*

| Reagent type (species) or resource | Designation | Source or reference | Identifiers | Additional information |
|---|---|---|---|---|
| sequence-based reagent | oCYL363 | This study | | GCACAAACCTGGAGATCG; Forward PCR primer for *lin-15B* locus. 200 bp upstream of n744 mutation. |
| sequence-based reagent | oCYL374 | This study | | AGAAGATGATGATTATGAGGAGG; Forward PCR primer for *lin-35(n745)* locus. 395 bp up stream of n745 mutation. |
| sequence-based reagent | oCYL375 | This study | | GAAGAAGCAGCAGAGTAAATTC; Reverse PCR primer for *lin-35(n745)* locus. 276 bp down stream n745 mutation. |
| sequence-based reagent | oCYL402 | This study | | TGGAGACTACAAATCCCACAG; Forward PCR primer for *dpl-1 (n3643)* locus, 270 bp up stream of n3643 site. |
| sequence-based reagent | oCYL405 | This study | | GTACGTAATATCGTTTGGTAACGG; Reverse PCR primer for *dpl-1(n3643)* locus, 270 bp down stream of n3643 site. |
| sequence-based reagent | oCYL668 | This study | | Repair ssODN for nos-2 deletion. See Supplementary file S10 for sequence information. |
| sequence-based reagent | oCYL977 | This study | | Repair ssODN for mes-3 C'ter OLLAS tag. See Supplementary file S10 for sequence information. |
| sequence-based reagent | gBLOCK4 | This study | | First 138nt is gpd-2/3 outron followed by the sequence of recoded first 20 amino acid of LIN-15B. See Supplementary file S10 for sequence information. |
| recombinant DNA reagent | pSL270 | This study | | contains *GFP-H2B::tbb2 3'UTR* from pCFJ420 and *gpd-2/3* outron plus 60 nt *lin-15B* 5' sequence from gBLOCK4 |
| software, algorithm | Diffbind | DOI: 10.18129/B9.bioc.DiffBind | RRID:SCR_012918 | |
| software, algorithm | hisat2 | DOI: 10.1038/nprot.2016.095 | RRID:SCR_015530 | |
| software, algorithm | htseq-count | DOI: 10.1093/bioinformatics/btu638 | RRID:SCR_011867 | |
| software, algorithm | cuffdiff | http://cole-trapnell-lab.github.io/cufflinks/ | RRID:SCR_001647 | |
| software, algorithm | Slidebook 6 | https://www.intelligent-imaging.com/slidebook | RRID:SCR_014300 | |

## Worm handling, RNAi, sterility counts

*C. elegans* was cultured according to standard methods (*Brenner, 1974*). RNAi knockdown experiments were performed by feeding on HT115 bacteria (*Timmons and Fire, 1998*). Feeding constructs were obtained from Ahringer or OpenBiosystem libraries or PCR fragments cloned into pL4440. The empty pL4440 vector was used as negative control. Bacteria were grown at 37°C in LB +ampicillin

(100 µg/mL) media for 5–6 hr, induced with 5 mM IPTG for 30 min, plated on NNGM (nematode nutritional growth media)+ampicillin (100 µg/mL)+IPTG (1 mM) plates, and grown overnight at room temperature. Embryos isolated by bleaching gravid hermaphrodites, or synchronized L1s hatched in M9, were put onto RNAi plates. For sterility counts, the progeny of at least six gravid adult hermaphrodites were tested. Adult progenies were scored for empty uteri ('white sterile' phenotype) on a dissecting microscope. For all immunostaining and smFISH experiments shown in *Figures 2D*, *6A and B*, *Figure 2—figure supplement 2*, *Figure 4—figure supplement 1* and *Figure 6—figure supplement 1*, worms were grown at 25°C. For live embryo imaging and synMuvB related experiments shown in *Figure 5*, *Figure 5-figure supplement 1*, *Figure 6C and D* and *Figure 4—figure supplement 1C*, worms were grown at 20°C.

To verify the efficiency of RNAi treatments used to create sequencing libraries, we scored animals exposed to the same RNAi feeding conditions for maternal-effect sterility. For *nos-1(gv5)* strain on *nos-2* RNAi, sterility was 81 ± 10% at 20°C and 86 ± 6% at 25°C; *mes-2(RNAi)* maternal effect sterility was 51 ± 1.4% and *mes-4(RNAi)* maternal effect sterility was 95.5 ± 3.5%. To test the efficiency of the double RNAi treatment for *nos-1(gv5)nos-2(RNAi);lin15B(RNAi)* RNA-seq libraries, we performed two additional controls. First we exposed a *nos-2::FLAG* strain (*Paix et al., 2014*) to the same RNAi feeding conditions and stained the embryos with α-FLAG antibody to confirm knock down of *nos-2* (4/15 embryos showed weak staining, compared to 15/15 embryos with strong staining in the untreated controls). Second, we exposed a *lin-15B::GFP* strain (*Paix et al., 2014*) to the same double RNAi feeding conditions and observed no GFP expression in embryos. *nos-1(gv5)nos-2(RNAi);lin15B (RNAi)* animals gave 34 ± 19% sterile progenies.

## Generation of *nos-2* null allele by CRISPR-mediated genome editing

See *Supplementary files 6* (CRISPR/Cas9 strain table) and key resources table for lists of strains and CRISPR reagents. The *nos-2(ok230)* allele removes the *nos-2* coding region and a flanking exon in the essential gene *him-14*, resulting in embryonic lethality. To create a *nos-2* null allele that does not affect *him-14* function, we deleted the *nos-2* open reading frame using CRISPR/Cas9-mediated genome editing (*Paix et al., 2015*). Consistent with previous reports (*Mainpal et al., 2015*; *Subramaniam and Seydoux, 1999*), *nos-2(ax3103)* animals are viable and fertile and *nos-1(gv5)nos-2(ax3103)* double mutants are maternal effect sterile (*Figure 5A*).

## Immunostaining

Adult worms were placed on 3-wells painted slides in M9 solution (Erie Scientific co.) and squashed under a coverslip to extrude embryos. Slides were frozen by laying on pre-chilled aluminum blocks for >10 min. Embryos were permeabilized by freeze-cracking (removal of coverslips from slides) followed by incubation in methanol at −20°C for 15 min, and then in pre-chilled acetone at −20°C for 10 min. Slides were blocked twice in PBS-Tween (0.1%)-BSA (0.1%) for 15 min at room temperature, and incubated with 75 µl primary antibody overnight at 4°C in a humidity chamber. Antibody dilutions (in PBST/BSA): Rabbit α-LIN-15B 1:20,000 (SDQ3183, gift from Dr. Susan Strome), Rabbit α-MES-4 1:400 (Gift from Dr. Susan Strome), mouse K76 1:10 (DSHB), Rat α-OLLAS-L2 1:200 (Novus Biological Littleton, CO), Rat α-OLLAS 1:80 (Gift from Dr. Jeremy Nathans), mouse α-FLAG M2 1:500 (Sigma F3165). Secondary antibodies (Molecular Probes/Thermo Fisher Sci.) were applied for 1 ∼ 2 hr at room temperature. MES-3 was tagged with the OLLAS epitope at the C-terminus using CRISPR genome editing (*Paix et al., 2015*).

## Confocal microscopy

Fluorescence microscopy was performed using a Zeiss Axio Imager with a Yokogawa spinning-disc confocal scanner. Images were taken and stored using Slidebook v6.0 software (Intelligent Imaging Innovations) using a 40x or 63x objective. Embryos were staged by DAPI-stained nuclei in optical Z-sections and multiple Z-sections were taken to include germ cells marked by α-PGL-1 (K76) staining. For images of embryonic PGCs, a single Z-section was extracted at a plane with the widest area of DAPI staining for nuclear signal of LIN-15B, MES-3, and MES-4. For MES-2-GFP, the Z-section was determined based on widest area of GFP signal. Equally normalized images were first taken by Slidebook v6.0, and contrasts of images were equally adjusted between control and experimental sets using Image J.

## Germ cell isolation and sorting

RNAi treatments for sorting experiments were done by seeding synchronized L1 (hatched from embryos incubated in M9 overnight) onto RNAi plates and growing them to gravid adults. Additional RNAi or control bacteria were added once to ensure enough food to support development. Early embryos were harvested from gravid adults. These embryos were either used directly to isolate embryonic PGCs or incubated for 12 ~ 16 hr in M9 solution until reaching the L1 stage for PGCs isolation. To isolate L1 PGCs from fed animals, the L1s were plated onto RNAi plates for additional 5 hr before processing for PGC isolation. For RNA-seq experiments described in *Figure 1* and *Figure 2*, RNAi treatments were done at 25°C. For the rest of RNA-seq experiments, RNAi treatments were done at 20°C. See *Supplementary file 7* for sequencing library information.

To isolate PGCs from embryos, cell dissociation was performed as described in *Strange et al. (2007)* (*Strange et al., 2007*) with the following modifications: $10^6$ embryos were treated in 500 µl chitinase solution (4.2 unit of chitinase (Sigma # C6137) in 1 ml of conditioned egg buffer). After chitinase treatment, embryos were collected by centrifugation at ~900 xg for 4 mins at 4°C and resuspended in 500 µl accumix-egg buffer solution for dissociation (Innovative Cell Techologies, AM105, 1:3 dilution ratio in egg buffer). In the final step, cells were resuspended in chilled egg buffer before sorting using BD FACSArialI. 65,000 ~ 120,000 PGL-1::GFP PGCs were used for RNA isolation.

To isolate PGCs from L1 larvae, total of >5 million L1 divided into ~500,000 L1 per reaction were used for cell dissociation as described in Zhang and Kuhn (*Zhang and Kuhn, 2013*) (www.worm-book.org/chapters/www_cellculture/cellculture.html#sec6-2) with the following modifications: starved and fed (for 5 hr) L1 were incubated with freshly thawed SDS-DTT solution for 2 min and 3 min, respectively, with gentle agitation using a 1000 µl pipette tip. Pronase treatment was performed using 150 µl of 15 mg/ml pronase (Sigma P6911). Pronase treatment was stopped by adding 1000 µl conditioned L-15 medium and spin at 1600xg for 6 min. Cells were resuspended in chilled egg buffer and washed three times to remove debris before sorting using BD FACSArialI or Beckman Coulter MoFlo sorter. ~75,000 sorted cells were pelleted at 1600xg for 5 min, snap frozen and saved in −80°C for later RNAseq analysis.

To assay the purity of isolated PGCs, aliquot of sorted PGCs were either passed through FACS sorter again to re-analyzed their GFP expression or subjected to GFP positive cell counting under microscope. PGC purity is defined by the percentage of GFP positive and propidium iodide negative in the sorted population. The purity of sorted embryonic PGCs is 95.7 ± 3.8% (N = 3); The purity of sorted L1 PGCs is 94.7 ± 4.7% (N = 10). From purified embryonic cells, we identified 1347 PGC enriched genes (enrichment over somatic blastomeres). We cross-reference our embryonic PGC enriched gene set with other published PGC or germline enriched gene sets. 392/1347 embryonic PGC enriched genes were identified as PGC enriched genes in *Spencer et al., 2011* (in which 979 genes with enriched expression in Z2/Z3); 700/1347 were characterized as either germline specific or germline enriched genes in *Gaydos et al., 2012*. The result is summarized in *Supplementary file 8*. The reproducibility of sorting/RNAseq procedure is demonstrated by PCA analysis as described in the section of RNAseq library preparation and analysis.

## RNA extraction

RNA was extracted from sorted cells using TRIZOL. The aqueous phase was transferred to Zymo-SpinTM IC Column (Zymo research R1013) for concentration and DNase I treatment as described in manual. RNA quality was assayed by Agilent Bioanalyzer using Agilent RNA 6000 Pico Chip. All RNAs used for library preparation had RIN (RNA integrity number)>8.

## RNAseq library preparation and analysis

Three different RNA-seq library preparation methods were used for this study: SMART-seq, which uses poly-A selection (*Figures 1* and *2*), NuGEN Ovation, which uses random priming (*Figure 2—figure supplement 1*, top), and Truseq combined with Ribozero to remove ribosomal RNAs (all other figures). The first two methods allow library construction from <10 ng of total RNA, whereas the later method requires >50 ng total RNA. We compared SMART-seq and Truseq-Ribozero performance on L1 PGCs isolated from wild-type and *nos-1(gv5)nos-2(RNAi)* and observed identical trends, with an overall higher number of misregulated genes identified with Truseq-Ribozero (Compare *Figure 1* (SMART-seq) and *Figure 1—figure supplement 1B, C and D* (Truseq/Ribozero). For

the experiment shown in *Figure 2—figure supplement 1* (top panels) where we compared RNA levels between embryonic PGCs and an oocyte library reference, we used Nugen Ovation libraries which avoids any bias due to poly-A selection while allowing library construction from <3 ng of RNA. For all experiments, control and experimental libraries were made using the same method. *Supplementary file 5* contains lists of misregulated genes from analyses. *Supplementary file 7* lists all the RNA-seq libraries used in this study and the corresponding figures.

SMART-seq libraries: libraries were made from 2 ng of total RNA isolated from sorted PGCs from worms grown at 25°C. Libraries were constructed using SMART-seq v4 Ultra Low input RNA kit (Clontech, Cat. No. 634888) followed by Low Input Library Prep Kit (Clontech, Cat. No. 634947). The cDNAs were then fragmented using Covaris AFA system at the Johns Hopkins University microarray core and cloned using the Low Input library prep Kit.

NuGEN Ovation libraries: libraries were made from 3 ng of total RNA isolated from sorted cells from worms grown at 25°C. Libraries were constructed using Nugen Ovation system V2 (#7102–08) followed by Nugen Ultralow library system.

TruSeq libraries: 50 ng of total RNA isolated from sorted PGCs from L1 worms grown at 20°C were subjected to Ribozero kit (illumina, MRZE706) to remove rRNA. Libraries were constructed using Truseq Library Prep Kit V2.

All cDNA libraries were sequenced using the Illumina Hiseq2000/2500 platform. Differential expression analysis was done using Tophat (V.2.0.8) and Cufflink (V.2.0.2). Cuffdiff accepts multiple biological replicates and uses Benjamini–Hochberg multiple hypothesis to compute false discovery rate (FDR). The cutoff of FDR(q value)=0.05 was used as a significance cutoff for all the analyses in this study. The command lines for Tuxedo suit are listed as below:

For each biological sample, sequencing reads were first mapped to ce10 reference genome using tophat2:

*$ tophat2 -p 12 g 1 –output-dir segment-length 20 –min-intron-length 10 –max-intron-length 25000 G < gene.gtf> –transcriptome-index<Name.fastq>*

For differential gene expression analysis, sets of independent mutant and control mapped reads (e.g biological replicates) were used in cuffdiff analysis:

*$ cuffdiff -p 12 -o < output > compatible-hits-norm –upper-quartile-norm -b < genome.fa> <genes.gtf> <tophat output_sample 1, tophat output_sample 2, tophat output_sample 3,..> <tophat output_control1, tophat output_control2, tophat output_control3,.. >*

Gene set enrichment analysis for four different categories and correlation of gene expression were done using R functions. R function *intersect()* was used to extract overlapping lists. Plots were drawn using R package and Prism software.

For correlation plots of gene expression shown in *Figure 4C–F*, information from different pairs of cuffdiff analyses (WT vs *mes-2*, WT vs *mes-4* and WT vs *nos-1/2*) was used. Genes with sufficient aligned reads to pass statistical test (OK status in test status from cuffdiff output) were kept, and those without enough alignments (NOTEST, LOWDATA in test status), or other conditions prevent statistical testing were excluded. Values of Log2 fold change were extracted from each cuffdiff output file and list of genes were further consolidated to generated correlation plots. The data process results in different number of genes in selected categories (1173 vs 1250 in X-linked genes, and 1063 vs 1092 in autosomal oocyte genes). However, majority of genes were overlapped between comparisons (1117 for X-linked genes and 1062 for autosomal oocyte genes)

In *Figure 6F*, the area-proportional Venn diagram was created using the VennDiagram R package. For comparisons shown in *Figure 2—figure supplement 1A*, oocyte transcriptome data was extracted from *Stoeckius et al. (2014)*, and embryonic soma and germ cells expression profiles were from this study (*Supplementary file 7*). Expression of each gene was log10 transformed, ranked and ordered. Correlations were plotted using custom R codes and can be found in *Figure 2—figure supplement 1A* source code.

## Principal component analysis

Principal component Analysis (PCA) was used to evaluate reproducibility of RNA-seq experiments. PCA revealed clustering of biological replicates with the same library preparation procedure as shown in *Figure 2—figure supplement 3*. In *Figure 2—figure supplement 3A*, two different sets of libraries (one set was made with NuGEN protocol and the other was made with SMART-seq protocol) were generated using the same RNA and clustered differently, suggesting different library

making procedures could introduce biases. Sequence reads were mapped to transcriptome version ce10 using Hisat2. HTseq-count was used to generate raw counts for each gene. The command lines are listed as below.

*$hisat2 -x < hisat2-index> -S < output file> -q < iinput file> –known-splicesite-infile<elegans_splicesites.txt> –no-softclip*

*$htseq-count -s no <genes.gtf> > outputfile.genecount*

The gene count information from HTseq-count (*Supplementary file 9*) was subject to regularized log transformation (rlog) and plotPCA in DEseq2 package.

## Gene categories

We defined four gene categories based on expression characteristics reported in published microarray, serial analysis of gene expression (SAGE), and RNAseq data sets that profiled specific tissues or whole worms with or without a germline (*Gaydos et al., 2012*; *Meissner et al., 2009*; *Ortiz et al., 2014*; *Reinke et al., 2004*; *Wang et al., 2009*). The oocyte category (1594 genes) and sperm category (2042 genes) are genes with differential enrichment in dissected female gonads from adult *fog2(q71)* animal compared to dissected male gonads from adult *fem-3(q96)* animals (*Ortiz et al., 2014*). The soma category (2684 genes) was obtained by taking genes with SAGE tags in at least one somatic tissue (intestine, muscle, or nerve) as described in *Gaydos et al., 2012*, and substracting from that list all genes in the oocyte and sperm categories described above. The pregamete category (1694 genes) was constructed by adding the germline-enriched and the germline-specific gene sets from *Gaydos et al., 2012* and substracting from that list all genes in the oocyte and sperm categories described above. Germline-enriched genes include genes whose expression is significantly higher in germline based on comparison of adults with and without a germline (*Reinke et al., 2004*). Germline-specific genes are those with SAGE tags in dissected germlines and not in somatic tissues (intestine, muscle and nerve cells). For gene sets enrichment test, we used total number of 15851 expressed genes with RPKM >0.1 as the cutoff from our PGC RNA-seq experiments to calculate 'expected' values for each category.

Expected value = (No. of significantly changed genes) x (No. of genes in category/15851). Hypergeomatric test was performed to derived p-values (hypergeometric probability), and listed in figure legends.

## ATAC-seq library preparation and analysis

ATAC-seq was performed as described in *Buenrostro et al., 2015*. Experimental pipeline was described as follows:

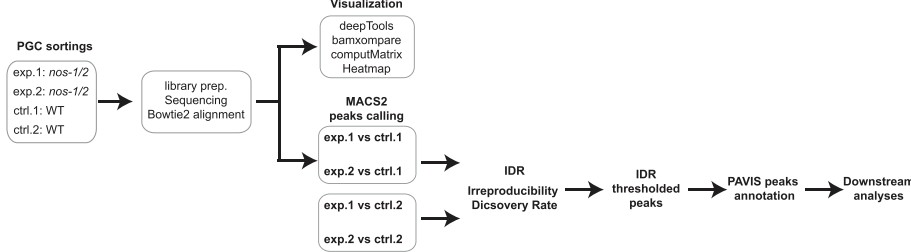

**Scheme 1.** Experimental procedure for ATAC-seq analysis.
DOI: https://doi.org/10.7554/eLife.30201.024

30,000 sorted L1 PGCs were washed with 60 µl cold cell culture grade PBS once and spun at 2000xg for 10 min. Cell nuclei were isolated by resuspending cell pellets in cold lysis buffer (10 mM Tris-Cl pH7.4, 10 mM NaCl, 3 mM MgCl$_2$, 0.1% Igepal CA-630) followed by centrifugation at 3500xg for 10 min at 4°C. The transposition reaction was performed with a 50 µl reaction mixture (25 µl TD, 2.5 µl TDE, 22.5 µl nuclease-free H$_2$O. Illumina, Nextera DNA library preparation Kit FC-121–1030) at 37°C for 30 min. Transposed DNA was purified using Qiagen MinElute kit and saved in −20°C. qPCR was used to determine appropriate PCR cycle number for PCR amplification as detailed in Buenrostro et al. 6–7 cycles of PCR amplification were used. Final cDNA libraries (150 bp to 700 bp) were selected using Agencourt AMPure beads (Beckman-Coulter A63880). Two biological samples for wild type and *nos-1(gv5)nos-2(RNAi)* were sequenced with Hiseq2500 platform.

Two biological replicates for control *and nos-1(gv5)nos-2(RNAi)* samples were independently mapped to *C. elegans* ce10 reference genome using bowtie2 (v2.1.0). Peaks from individual ATAC-seq sample were called using MACS2 packing with options -p 1e-3 –nomodel –shift −100 –extsize 200. To evaluate the correlation between two biological replicates, Diffbind package was then used to perform PCA analysis and RPKM for peaks were extracted from matadata using function dba.peakset(DBA object, bRetrieve = T, DataType = DBA_DATA_FRAME). For correlation plots, peaks with RPKM >1 were kept and subjected to log2 transformation and correlations for replicates were calculated using Pearson correlation (*Figure 3—figure supplement 2*).

To identify locus with *nos-1/2*-dependent features (peaks), mapped reads from wild-type were used as reference sample and the *callpeak* function in MACS2 package was used as described below:

$Macs2 callpeak -t [nos_rep1.sam] -c [con_rep1.sam] –outdir –f SAM –g ce –n **exp1_vs_reference1** –p 0.01 –to-large

$Macs2 callpeak -t [nos_rep2.sam] -c [con_rep1.sam] –outdir –f SAM –g ce –n **exp2_vs_reference1** –p 0.01 –to-large

$Macs2 callpeak -t [nos_rep1.sam] -c [con_rep2.sam] –outdir –f SAM –g ce –n **exp1_vs_reference2** –p 0.01 –to-large

$Macs2 callpeak -t [nos_rep2.sam] -c [con_rep2.sam] –outdir –f SAM –g ce –n **exp2_vs_reference2** –p 0.01 –to-large

To identify nos-1/2-dependent feature (peaks) with confidence, we followed the principle of ENCODE Irreproducibility Discovery Rate (IDR) framework as described in https://sites.google.com/site/anshulkundaje/projects/idr#TOC-FLAGGING-REPLICATES-FOR-LOW-CONSISTENCY. For IDR analysis, pairwise consistency analysis was done on replicate peak files as described below:

$Rscript batch-consistency-analysis.r **[exp1_vs_refernece1_** peakfile] **[exp2_vs_reference1_** peakfile] −1 [output_**set1**.perfix] 0 F p.value

$Rscript batch-consistency-analysis.r **[exp1 vs reference2_** peakfile] **[exp2_vs_reference2_** peakfile] −1 [output_**set2**.perfix] 0 F p.value

To obtain a list of overlapped peaks between replicates, IDR cutoff was set to 0.1. 1414 peaks were selected based on IDR cutoff and peaks were annotated using PAVIS (https://manticore.niehs.nih.gov/pavis2/). At the end, 221 peaks with location at upstream region of genes were extracted and gene IDs were cross-referenced with RNA-seq analysis for downstream analysis.

To plot heatmap for ATAC-seq analysis, *bamCompare* and *computeMatrix* in deepTools package (http://deeptools.readthedocs.io/en/latest/) were used to visualize merged ATAC-seq profile of *nos-1/2*-dependent genes as shown in *Figure 3—figure supplement 1*. ATAC-seq reads from replicates were merged and mapped to *C. elegans* ce10 reference genome using bowtie2 (v2.1.0). Command lines were listed as below:

$bamCompare -b1 < nos-1/2.bam > -b2 < wild type.bam> -o < Name1.bw> –ratio ratio –normalizeUsingRPKM -ignore chrM -bs 10 p max/2

$computeMatrix reference-point –referencePoint TSS -b 2000 -a 2000 R < nos-1/2- dependent_gene.bed> -S < Name1.bw> -o < Name2.gz> –sortUsing max – skipZeros -bs 10 p 2

$plotHeatmap -m < Name2.gz> –zMin 0 –colorList –heatmapHeight 20 – heatmapWidth 5 -out < heatmap .png>

## Quantitative RT-PCR assay

To verify our analysis pipeline for RNAseq data, quantitative RT-PCR (qRT-PCR) reactions using sequencing libraries as templates were performed. The cDNA libraries were diluted to 1 nM before performing qRT-PCR. Primers for qRT-PCR were listed in key resources table. Enrichment of target mRNAs between wild-type and *nos-1/2* was calculated using ΔΔCt with *tbb-2* expression then normalized to wild-type control. Fold changes were plotted and significance was calculated by paired t-test.

## Technical v biological replicates

Biological replicates refer to experiments performed on independently treated hermaphrodites (in the case of RNA-seq libraries, this refers to worms exposed to independent RNAi treatments

followed by cell sorting and RNA extraction). All in vivo technical replicates refer to observations in the same strain from separate zygotes.

## Datasets

Datasets generated in this paper are available at GEO accession GSE100651 for ATAC-seq and GSE100652 for RNA-seq.

# Additional information

### Funding

| Funder | Grant reference number | Author |
|---|---|---|
| National Institutes of Health | R01HD37047 | Geraldine Seydoux |
| Howard Hughes Medical Institute | | Geraldine Seydoux |
| Damon Runyon Cancer Research Foundation | DRG-2417-13 | Chih-Yung Sean Lee |

The funders had no role in study design, data collection and interpretation, or the decision to submit the work for publication.

### Author contributions

Chih-Yung Sean Lee, Conceptualization, Formal analysis, Funding acquisition, Investigation, Methodology, Writing—original draft, Writing—review and editing; Tu Lu, Data curation, Software, Investigation; Geraldine Seydoux, Conceptualization, Funding acquisition, Writing—original draft, Writing—review and editing

### Author ORCIDs

Chih-Yung Sean Lee (iD) http://orcid.org/0000-0003-0982-2150
Tu Lu (iD) http://orcid.org/0000-0002-5697-300X
Geraldine Seydoux (iD) http://orcid.org/0000-0001-8257-0493

### Decision letter and Author response

Decision letter https://doi.org/10.7554/eLife.30201.043
Author response https://doi.org/10.7554/eLife.30201.044

# Additional files

### Supplementary files

• Supplementary file 1. Gene categories. These categories were based on previously published data sets as described in methods section.
DOI: https://doi.org/10.7554/eLife.30201.025

• Supplementary file 2. Genes with *nos-1nos-2*-dependent chromatin features. Excel sheet 1. List of 221 genes that acquired open chromatin in *nos-1(gv5)nos-2(RNAi)* L1 PGCs. Excel sheet 2. List of 29 genes that acquired close chromatin in *nos-1(gv5)nos-2(RNAi)* L1 PGCs.
DOI: https://doi.org/10.7554/eLife.30201.026

• Supplementary file 3. Average gene expression level between x-linked and autosomal genes.
DOI: https://doi.org/10.7554/eLife.30201.027

• Supplementary file 4. Expression level of selected genes.
DOI: https://doi.org/10.7554/eLife.30201.028

• Supplementary file 5. Lists of differential expression analyses. Data sheets contain significantly up- and downregulated genes from pairwise comparisons using ciffdiff as described in the methods section. FPKM values were extracted from differential expression analysis between wild-type and nos-1 (gv5)nos-2(RNAi) PGCs using SMART-seq libraries.

DOI: https://doi.org/10.7554/eLife.30201.029

• Supplementary file 6. Information for strains generated by CRISPR/cas9.
DOI: https://doi.org/10.7554/eLife.30201.030

• Supplementary file 7. Information for sequencing libraries.
DOI: https://doi.org/10.7554/eLife.30201.031

• Supplementary file 8. List of embryonic Z2/Z3 enriched genes. From purified embryonic cells, we identified 1347 PGC enriched genes (enrichment over somatic blastomeres). In this table, we cross-referenced our embryonic PGC enriched gene set with other published PGC or germline enriched gene sets. 392/1347 embryonic PGC enriched genes were identified as PGC enriched genes in Spencer et al., 2011- *Supplementary file 2* (in which 979 genes with enriched expression in Z2/Z3); 700/1347 were characterized as either germline specific or germline enriched genes in *Gaydos et al., 2012*.
DOI: https://doi.org/10.7554/eLife.30201.032

• Supplementary file 9. Gene count tables for PCA plots. Sheet 1. A table contains HTseq-count output using all Truseq libraries. Sheet 2. A table contains HTseq-count output using SMART-seq libraries.
DOI: https://doi.org/10.7554/eLife.30201.033

• Supplementary file 10. Additional sequence information for oligos.
DOI: https://doi.org/10.7554/eLife.30201.034

• Transparent reporting form
DOI: https://doi.org/10.7554/eLife.30201.035

### Major datasets

The following datasets were generated:

| Author(s) | Year | Dataset title | Dataset URL | Database, license, and accessibility information |
| --- | --- | --- | --- | --- |
| Lee CY, Lu T, Seydoux G | 2017 | Genome-wide maps of chromatin structure in wild-type and nanos mutant primordial germ cells in C. elegans. | https://www.ncbi.nlm.nih.gov/geo/query/acc.cgi?acc=GSE100651 | Publicly available at the NCBI Gene Expression Omnibus (accession no. GSE100651) |
| Lee CY, Lu T, Seydoux G | 2017 | Chromatin reprogramming in primordial germ cells requires Nanos-dependent down-regulation of the synMuvB transcription factor LIN-15B | https://www.ncbi.nlm.nih.gov/geo/query/acc.cgi?acc=GSE100652 | Publicly available at the NCBI Gene Expression Omnibus (accession no.GSE100652) |

The following previously published dataset was used:

| Author(s) | Year | Dataset title | Dataset URL | Database, license, and accessibility information |
| --- | --- | --- | --- | --- |
| Ortiz MA, Noble D, Sorokin EP, Kimble J | 2014 | A New Dataset of Spermatogenic vs. Oogenic Transcriptomes in the Nematode Caenorhabditis elegans | http://www.g3journal.org/highwire/filestream/472440/field_highwire_adjunct_files/0/TableS1.xlsx | Supplemental Table S1 |

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
