## [Decision Letter]

Thank you for submitting your article "Specification of the germline by Nanos-dependent down-regulation of the somatic synMuvB transcription factor LIN-15B" for consideration by *eLife*. Your article has been reviewed by three peer reviewers, and the evaluation has been overseen by Julie Ahringer as Reviewing Editor and Marianne Bronner as the Senior Editor. The reviewers have opted to remain anonymous.

The reviewers have discussed the reviews with one another and the Reviewing Editor has drafted this decision to help you prepare a revised submission.

Summary:

This paper investigates how Nanos proteins (NOS) control the unique gene expression and development programs of primordial germ cells (PGCs) in *C. elegans*. The authors provide evidence that NOS-1 and NOS-2 promote activation of pre-gametic PGC genes, suppress premature expression of oogenic germline genes and some somatic genes, and promote degradation of many maternal mRNAs. Interestingly, this regulation is at both post-transcriptional and transcriptional levels. Remarkably, the authors also provide compelling evidence that most of the functions of Nanos in germline establishment and function in *C. elegans* center on its regulation of DRM-class transcriptional regulators, especially LIN-15B, in part to suppress chromatin modifiers (PRC2 and NSD methyl transferases; MES proteins in worms).

Overall, the paper is well conceived and written, and the presented data broadens our understanding of Nanos function in germline development. The experiments are nicely designed and the results consistent and generally support the conclusions drawn. There are, however, some issues and questions that should be addressed to make the manuscript easier to interpret, as well as some conclusions and statements that need clarification.

Essential revisions:

1) The data strongly support the idea that *lin-15b* repression is a major function of NOS-1 and -2. The authors argue that this allows PRC2/MES methyl transferases to silence oogenic and somatic gene genes in PGCs. While some results here and earlier work support this idea, the data here seem to suggest that NOS and LIN-15b may also act independently of PRC2/MES in PGCs, which could be important for NOS functions. First, autosomal oogenic genes regulated by NOS do not correlate well with autosomal oogenic genes controlled by MES-2 and -4 (Figure 4), while MES-2 and MES-4 autosomal oogenic targets correlate strongly with each other (Figure 4). Second, although NOS-dependent X chromosome genes overlap well with MES-dependent genes on X, correlation is weaker than MES genes with each other (Figure 4), suggesting that even some X targets may be NOS-specific. Third, authors do not investigate the extent to which the strong *lin-15b*(lf) suppression of *nos-1;nos-2*(lf) depends on MES function. Considering *lin-15b*-dependent and mes-dependent genes in Figure 4, Figure 6, and Supplementary file 5: Does *lin-15b* loss suppress misregulation of autosomal oogenic genes or the increased perdurance of putative maternal mRNAs (from Figure 2) in *nos* mutants? Do mes-sensitive gene sets generally lack these kinds of oocyte genes? If the data support it, authors should discuss a potential MES-independent pathway controlled by NOS proteins and LIN-15B (in Results, model, and Discussion).

2) How was purification and quality of PGCs assessed after FACS isolation? Authors should include either a supplemental figure, or state stats or other measures. Relatedly, it would help to more explicitly comment on specificity of PGC gene expression profiles relative to known PGC mRNAs and RNA Seq data from embryo somatic cells or whole L1s (e.g. are most known PGC mRNAs enriched and somatic genes depleted in PGC data relative to somatic cell or L1 RNA Seq?). Figure 2—figure supplement 1 seems to partly support PGC specificity.

3) Additional information on RNA Seq data analysis is needed. Pearson correlations or other measures of biological replicates need to be stated or shown, to address reproducibility. For volcano plots, how were q values determined? Were biological or technical replicates used in the Cuffdiff analysis, or were q values derived from distributions and assumptions within a single dataset pair? Were the RNAseq datasets used as independent inputs to the DE software (and the reads not merged prior to analysis)? This should be made more explicit in the methods. Please also add a column to Supplementary file 8 giving the number of reads obtained for each dataset.

4) Could the authors please clarify how the expected number of genes (per stage) was calculated or obtained (Figure 1, Figure 2 and Figure 3). The p values for "observed" versus "expected" gene numbers should be determined and added to figures (asterisks would suffice). Second, assuming "expected" reflects the fraction of genes in a category among all expressed genes, please provide these total expressed gene numbers for the datasets (somewhere). What was rpkm cutoff used to define a gene as expressed in a sample (in Materials and methods)?

5) Results Figure 2. In WT embs vs. L1s, 45 gamete genes go up in the transition and few if any go down. In the *nos* mutant, ~110 pregamete genes go down in the transition, and a more go up in than in WT. This seems to be a major change that is not commented on- and should be. Does this represent a major transcriptional shift in addition to the proposed shift in mRNA turnover? Are these changes minor or dramatic (by RPKM)?

6) Results in Figure 3. The ATAC seek shows that a significant # of *ooc* genes that have increased L1 PGC abundance in the *nos* mutant have what looks like an increased transcriptional response. This seems in contrast to the results in Figure 2, which are interpreted to represent increased mRNA stability in the mutants. Are these the same genes? If so, please explain what appears to be conflicting interpretations of why *ooc* genes are more abundant in the mutant PGCs.

7) Results in Figure 4. The data shows that more autosomal linked *ooc* mRNAs are increased in the *nos* mutant than in *mes* mutants, and the authors conclude that this difference is due to defective degradation of *ooc* genes in the *nos* mutant. Assuming the *mes* mutants exhibit defective transcriptional regulation, then there should be some X-linked *ooc* genes that are unaffected by loss of *mes* but are increased in the *nos* mutant. Is this so?

8) The first paragraph of the Discussion indicates that "Nanos is required to erase maternally-inherited somatic program". The results do not really show this – or if so, I missed it. Nanos seems to largely affect genes required for oogenesis, and if anything is being erased in the embryo to larval transition, it is the information that drove (or resulted from) maternal oogenesis that is likely inappropriate in pre-gamete germ cells and needs to be removed, particularly in hermaphrodite larvae which will initially engage male gametogenesis. The effect on somatic loci seems minor. Thus this claim thus seems overstated.

9) Although biological replicates were collected for ATAC-seq, according to the methods, the raw data were apparently merged prior to analysis, which would lead to an unequal contributions of the replicates. Additionally, with only one control and one experimental file, the p-values of peak calls are not meaningful. The four raw data files (two control and two experimental) should be used as input for peak calling, then reproducible peaks taken for example, as those in both sets of peak calls, or using IDR software as developed for ENCODE (https://www.encodeproject.org/software/idr/ https://www.encodeproject.org/software/idr/) or another method. Following this, signal tracks of the independent replicates (or the independent replicates relative to control) should be averaged and used for analysis. Please also provide information on the concordance of wild-type and nos mutant/RNAi ATAC-seq replicates (e.g., correlation of signal in peak regions) and improve the legend to Figure 3 (e.g., what are "accumulated ATAC-seq reads"?). Please add ATAC-seq library information to Supplementary file 8.

10) Please explain in the Materials and methods how the biological replicate RNAseq datasets were handled for the differential expression analyses (i.e., were they used as independent inputs to the DE software?) and add a column to Supplementary file 8 giving the number of reads obtained for each dataset.

11) Please define the term "pre-gametic." When and in what tissue would these genes normally be expressed and function? The gene list used for this indicates "pre-gamate: germline enriched and germline specific genes in Gaydos et al. – oogenic genes from Ortiz et al." Presumably this list would contain genes expressed in both oogenic and spermatogenic germ lines as well as sperm specific genes? (genes do occur on both pre-gametic and sperm lists in Supplementary file 1).

---

## [Author Response]

Essential revisions:1) The data strongly support the idea that lin-15b repression is a major function of NOS-1 and -2. The authors argue that this allows PRC2/MES methyl transferases to silence oogenic and somatic gene genes in PGCs. While some results here and earlier work support this idea, the data here seem to suggest that NOS and LIN-15b may also act independently of PRC2/MES in PGCs, which could be important for NOS functions. First, autosomal oogenic genes regulated by NOS do not correlate well with autosomal oogenic genes controlled by MES-2 and -4 (Figure 4), while MES-2 and MES-4 autosomal oogenic targets correlate strongly with each other (Figure 4). Second, although NOS-dependent X chromosome genes overlap well with MES-dependent genes on X, correlation is weaker than MES genes with each other (Figure 4), suggesting that even some X targets may be NOS-specific. Third, authors do not investigate the extent to which the strong lin-15b(lf) suppression of nos-1;nos-2(lf) depends on MES function. Considering lin-15b-dependent and mes-dependent genes in Figure 4, Figure 6, and Supplementary file 5: Does lin-15b loss suppress misregulation of autosomal oogenic genes or the increased perdurance of putative maternal mRNAs (from Figure 2) in nos mutants? Do mes-sensitive gene sets generally lack these kinds of oocyte genes? If the data support it, authors should discuss a potential MES-independent pathway controlled by NOS proteins and LIN-15B (in Results, model, and Discussion).

We thank the reviewers for their thorough analyses. As suggested, we have performed two new experiments to clarify the model.

1) Using in situ hybridization against two maternal mRNAs, we show that loss of *lin-15B* does *not* suppress the Nanos defect in maternal mRNA turnover (Figure 2—figure supplement 2). This confirms that defective maternal mRNA turnover is a defect intrinsic to Nanos PCGs that is not dependent on LIN-15B.

2) We demonstrate that MES function is still required for fertility in the *nos-1nos-2;lin-15B* triple mutant strain (Figure 5—figure supplement 3). This finding does not support a MES-independent pathway for fertility, and is consistent with the model that LIN-15B antagonizes MES function in PGCs, as it has been reported to do in somatic cells.

The new data support the following:

1) Nanos’s primary function is to promote the turnover of maternal mRNAs in PGCs.

2) Extended perdurance of maternal RNAs and proteins, including LIN-15B, in Nanos PGCs causes oocyte genes to become ectopically transcribed.

3) We suggest that this is due to LIN-15B antagonizing MES function, based on 1) previous data documenting such an antagonism in somatic cells, and 2) the observation that many of the genes upregulated in PGCs are also upregulated in PGCs lacking MES.

2) How was purification and quality of PGCs assessed after FACS isolation? Authors should include either a supplemental figure, or state stats or other measures. Relatedly, it would help to more explicitly comment on specificity of PGC gene expression profiles relative to known PGC mRNAs and RNA Seq data from embryo somatic cells or whole L1s (e.g. are most known PGC mRNAs enriched and somatic genes depleted in PGC data relative to somatic cell or L1 RNA Seq?). Figure 2—figure supplement 1 seems to partly support PGC specificity.

We performed several tests to assess the quality of our RNAseq data sets.

First, we examined the sorted cells by fluorescence microscopy for PGL-1::GFP expression (a PGC-specific marker) and determined that the sorted cells were 95.7 ± 3.8% PGCs (embryonic sort) and 94.7 ± 4.7% PGCs (L1 sort) (N= 10). See Materials and methods section Germ cell isolation and sorting.

Second, we used principal component analysis (PCA) to cluster our RNA-seq data sets and found that experimental replicates clustered together as expected (Figure 2—figure supplement 3).

Finally, we compared our data sets to others in the literature. First, we generated a list of PGC-enriched genes (1347 genes) by comparing the transcriptomes of sorted PGCs and sorted somatic blastomeres. We cross-referenced this list with published PGC or germline-enriched gene sets. We found that 392/1347 PGC-enriched genes were also identified as PGC-enriched genes in Spencer et al. 2011 (979 genes with enriched expression in Z2/Z3), and 700/1347 were characterized as either germline-specific or germline-enriched genes in Gaydos et al. 2012. By comparison only 61/1347 PGC-enriched genes were present in a soma-specific gene set (2684 genes) (Gaydos et al. 2012 and Supplementary file 9).

3) Additional information on RNA Seq data analysis is needed. Pearson correlations or other measures of biological replicates need to be stated or shown, to address reproducibility. For volcano plots, how were q values determined? Were biological or technical replicates used in the Cuffdiff analysis, or were q values derived from distributions and assumptions within a single dataset pair? Were the RNAseq datasets used as independent inputs to the DE software (and the reads not merged prior to analysis)? This should be made more explicit in the methods. Please also add a column to Supplementary file 8 giving the number of reads obtained for each dataset.

We have provided the information as requested:

PCA analysis: PCA plots are shown in Figure 2—figure supplement 3 and PCA command lines are included in a new section “Principle component analysis” in Materials and methods.

See Materials and methods section “RNAseq library preparation and analysis” for derivation of q values, handling of biological replicates, and command line usage.

We added a new column to Supplementary file 8 with number of mapped reads for each sequencing library.

4) Could the authors please clarify how the expected number of genes (per stage) was calculated or obtained (Figure 1, Figure 2 and Figure 3). The p values for "observed" versus "expected" gene numbers should be determined and added to figures (asterisks would suffice). Second, assuming "expected" reflects the fraction of genes in a category among all expressed genes, please provide these total expressed gene numbers for the datasets (somewhere). What was rpkm cutoff used to define a gene as expressed in a sample (in Materials and methods)?

We have added a new section “Gene categories” in Materials and methods. To calculate expected values, we used total number of 15851 expressed genes with RPKM > 0.1 as the cutoff from our PGC RNA-seq experiments.

Expected value = (No. of significantly changed genes) x (No. of genes in category/15851).

We performed Hypergeometric test to derive P values (hypergeometric probability) for each category and added asterisks to categories with P value lower than 0.01 or 0.001 as noted in the figure legends.

5) Results Figure 2. In WT embs vs. L1s, 45 gamete genes go up in the transition and few if any go down. In the nos mutant, ~110 pregamete genes go down in the transition, and a more go up in than in WT. This seems to be a major change that is not commented on- and should be. Does this represent a major transcriptional shift in addition to the proposed shift in mRNA turnover? Are these changes minor or dramatic (by RPKM)?

Yes, we believe that this difference is significant and cannot be explained solely by a delay in maternal mRNA turnover, since many oocyte transcripts appear to increase in abundance. The ATAC-seq data provide further support that transcription is affected. We have expanded our discussion of these observations in the subsection “PGCs lacking *nos-1* and *nos-2* activate the transcription of many genes normally silent in PGCs”.

We identify several maternally-provided transcription factors that, like LIN-15B, are also not turned over efficiently in Nanos PGCs, and speculate that the collective action of these factors may cause the complex gene upregulation observed in Nanos PGCs.

6) Results in Figure 3. The ATAC seek shows that a significant # of ooc genes that have increased L1 PGC abundance in the nos mutant have what looks like an increased transcriptional response. This seems in contrast to the results in Figure 2, which are interpreted to represent increased mRNA stability in the mutants. Are these the same genes? If so, please explain what appears to be conflicting interpretations of why ooc genes are more abundant in the mutant PGCs.

In the transcriptomic analyses, we cannot easily distinguish whether differences in RNA abundance are due to changes in mRNA turnover, de novo synthesis, or both. The ATAC-seq analysis gave us an independent method to assess the contribution of de novo synthesis. To assess the contribution of maternal mRNA turnover, we used transcriptomics of EMB PGCs (isolated before the onset of transcription) and in situ hybridization experiments using probes to known maternal mRNAs. We have rewritten significant portions of the Results section to make these analyses less confusing.

7) Results in Figure 4. The data shows that more autosomal linked ooc mRNAs are increased in the nos mutant than in mes mutants, and the authors conclude that this difference is due to defective degradation of ooc genes in the nos mutant. Assuming the mes mutants exhibit defective transcriptional regulation, then there should be some X-linked ooc genes that are unaffected by loss of mes but are increased in the nos mutant. Is this so?

Yes. We identified 267 up-regulated X-linked oogenic genes in *nos-1(gv5)nos-2(RNAi)* PGCs. Of those, 113 were *not* also upregulated in *mes-4*(RNAi) PGCs. We hypothesize that these transcripts correspond to maternal mRNAs that are not efficiently turned over in NOS PGCs.

8) The first paragraph of the Discussion indicates that "Nanos is required to erase maternally-inherited somatic program". The results do not really show this – or if so, I missed it. Nanos seems to largely affect genes required for oogenesis, and if anything is being erased in the embryo to larval transition, it is the information that drove (or resulted from) maternal oogenesis that is likely inappropriate in pre-gamete germ cells and needs to be removed, particularly in hermaphrodite larvae which will initially engage male gametogenesis. The effect on somatic loci seems minor. Thus this claim thus seems overstated.

We agree with the reviewer that this was misleading. We have clarified that what we define as “oocyte genes” → these include maternal transcripts that function to promote somatic cell fates in embryos. We have made this distinction clearer throughout the text.

9) Although biological replicates were collected for ATAC-seq, according to the methods, the raw data were apparently merged prior to analysis, which would lead to an unequal contributions of the replicates. Additionally, with only one control and one experimental file, the p-values of peak calls are not meaningful. The four raw data files (two control and two experimental) should be used as input for peak calling, then reproducible peaks taken for example, as those in both sets of peak calls, or using IDR software as developed for ENCODE (https://www.encodeproject.org/software/idr/ https://www.encodeproject.org/software/idr/) or another method. Following this, signal tracks of the independent replicates (or the independent replicates relative to control) should be averaged and used for analysis. Please also provide information on the concordance of wild-type and nos mutant/RNAi ATAC-seq replicates (e.g., correlation of signal in peak regions) and improve the legend to Figure 3 (e.g., what are "accumulated ATAC-seq reads"?). Please add ATAC-seq library information to Supplementary file 8.

As suggested, we have reanalyzed our ATAC-seq results (4 raw data files including two controls and two experimental sets) using IDR software as developed for the ENCODE project. The new analysis identified 221 genes with new ATAC-seq peaks in NOS PGCs compared to 247 genes in our previous analysis. This change does *not* alter our conclusions. We have replaced the old ATAC-seq results with this new analysis and included a description of the IDR procedure (Materials and methods) and correlation plot of signal in peak region between replicates in Figure 3—figure supplement 2.

10) Please explain in the Materials and methods how the biological replicate RNAseq datasets were handled for the differential expression analyses (i.e., were they used as independent inputs to the DE software?) and add a column to Supplementary file 8 giving the number of reads obtained for each dataset.

See answer in #4.

11) Please define the term "pre-gametic." When and in what tissue would these genes normally be expressed and function? The gene list used for this indicates "pre-gamate: germline enriched and germline specific genes in Gaydos et al. – oogenic genes from Ortiz et al." Presumably this list would contain genes expressed in both oogenic and spermatogenic germ lines as well as sperm specific genes? (genes do occur on both pre-gametic and sperm lists in Supplementary file 1).

Our “pregamete” gene list correspond to germline-enriched genes AND germline-specific genes (Gaydos et al.,2012) MINUS oocyte and sperm-specific genes (Ortiz et al. 2014). This pregamete gene list should contain genes expressed in the mitotic region of the germline, as well as in early meiotic germ cells (pachytene), in the context of both oogenic and spermatogenic germlines.